# An Exact Poly-Time Membership-Queries Algorithm for Extracting a Three-Layer ReLU Network

**Amit Daniely**
School of Computer Science and Engineering, The Hebrew University
and Google Research Tel-Aviv
amit.daniely@mail.huji.ac.il

**Elad Granot**
School of Computer Science and Engineering, The Hebrew University
elad.granot@mail.huji.ac.il

## Abstract

We consider the natural problem of learning a ReLU network from queries, which was recently remotivated by model extraction attacks. In this work, we present a polynomial-time algorithm that can learn a depth-two ReLU network from queries under mild general position assumptions. We also present a polynomial-time algorithm that, under mild general position assumptions, can learn a rich class of depth-three ReLU networks from queries. For instance, it can learn most networks where the number of first layer neurons is smaller than the dimension and the number of second layer neurons.

These two results substantially improve state-of-the-art: Until our work, polynomial-time algorithms were only shown to learn from queries depth-two networks under the assumption that either the underlying distribution is Gaussian (Chen et al. (2021)) or that the weights matrix rows are linearly independent (Milli et al. (2019)). For depth three or more, there were no known poly-time results.

## 1 Introduction

With the growth of neural-network-based applications, many commercial companies offer machine learning services, allowing public use of trained networks as a black-box. Those networks allow the user to query the model and, in some cases, return the exact output of the network to allow the users to reason about the model's output. Yet, the parameters of the model and its architecture are considered the companies' intellectual property, and they do not often wish to reveal it. Moreover, sometimes the training phase uses sensitive data, and as demonstrated in Zhang et al. (2020), inversion attacks can expose those sensitive data to one who has the trained model.

Nevertheless, the model is still vulnerable to membership query attacks even as a black box. A recent line of works (Tramer et al. (2016), Shi et al. (2017), Milli et al. (2019), Rolnick & Körding (2020), Carlini et al. (2020), Fornasier et al. (2021)) showed either empirically or theoretically that using a specific set of queries, one can reconstruct some hidden models. Theoretical work includes Chen et al. (2021) that proposed a novel algorithm that, under the Gaussian distribution, can approximate a two-layer model with ReLU activation in a guaranteed polynomial time and query complexity without any further assumptions on the parameters. Likewise, Milli et al. (2019) has shown how to exactly extract the parameters of depth-two networks, assuming that the weight matrix has independent rows (in particular, the number of neurons is at most the input dimension). Our work extends their work by showing:

1. A polynomial time and query complexity algorithm for **exact reconstruction** of a two-layer neural network with any number of hidden neurons, under mild general position assumptions; and

2. A polynomial time and a query complexity algorithm for exact reconstruction of a **three-layer neural network** under mild general position assumptions, with the additional assumptions that the number of first layer neurons is smaller than the input dimension and the assumption that the second layer has non-zero partial derivatives. The last assumption is valid for most networks with more second layer neurons than first layer neurons.

The mild general position assumptions are further explained in section 3.3. However, we note that the proposed algorithm will work on any two-layer neural network except for a set with a zero Lebesgue measure. Furthermore, it will work in polynomial time provided that the input weights are slightly perturbed (for instance, each weight is perturbed by adding a uniform number in $\left[-2^{-d}, 2^{-d}\right]$) At a very high level, the basis of our approach is to find points in which the linearity of the network breaks and extract neurons by recovering the affine transformations computed by the network near these points. This approach was taken by the previous theoretical papers Milli et al. (2019); Chen et al. (2021) and also in the empirical works of Carlini et al. (2020); Jagielski et al. (2019). In order to derive our results, we add several ideas to the existing techniques, including the ability to distinguish first from second layer neurons, which allows us to deal with three-layer networks, as well as the ability to reconstruct the neurons correctly in general depth-two networks with any finite width in a polynomial time, without assuming that the rows are independent.

## 2 RESULTS

We next describe our results. Our results will assume a general position assumption quantified by a parameter $\delta \in (0, 1)$, and a network that satisfies our assumption with parameter $\delta$ will be called $\delta$-*regular*. This assumption is defined in section 3.3. We note, however, that a slight perturbation of the network weights, say, adding to each weight a uniform number in $[-2^{-d}, 2^{-d}]$, guarantees that w.p. $1 - 2^{-d}$ the network will be $\delta$-regular with $\delta$ that is large enough to guarantee polynomial time complexity. Thus, $\delta$-regularity is argued to be a mild general position assumption. Throughout the paper, we denote by $Q$ the time it takes to make a single query.

### 2.1 DEPTH TWO NETWORKS

Consider a 2-layer network model given by

$$\mathcal{M}(\boldsymbol{x}) = \sum_{j=1}^{d_1} u_j \phi \left( \langle \boldsymbol{w}_j, \boldsymbol{x} \rangle + b_j \right) \tag{1}$$

where $\phi(x) = x^+ = \max(x, 0)$ is the ReLU function, and for any $j \in [d_1]$, $\boldsymbol{w}_j \in \mathbb{R}^d$, $b_j \in \mathbb{R}$, and $u_j \in \mathbb{R}$. We assume that the $\boldsymbol{w}_j$'s, the $b_j$'s and the $u_j$'s, along with the width $d_1$, are unknown to the user, which has only black box access to $\mathcal{M}(\boldsymbol{x})$, for any $\boldsymbol{x} \in \mathbb{R}^d$. We do not make any further assumptions on the network weights, rather than $\delta$-regularity.

**Theorem 1.** *There is an algorithm that given an oracle access to a $\delta$-regular network as in equation 1, reconstructs it using $O\left((d_1 \log(1/\delta) + d_1 d) Q + d^2 d_1\right)$ time and $O\left(d_1 \log(1/\delta) + d_1 d\right)$ queries.*

We note that by reconstruction we mean that the algorithm will find $d_1'$ and weights $\boldsymbol{w}_0', \ldots, \boldsymbol{w}_{d_1'}' \in \mathbb{R}^d, b_0', \ldots, b_{d_1'}' \in \mathbb{R}$, and $u_1', \ldots, u_{d_1'}' \in \mathbb{R}$ such that

$$\forall \boldsymbol{x} \in \mathbb{R}^d, \quad \mathcal{M}(\boldsymbol{x}) = \langle \boldsymbol{w}_0', \boldsymbol{x} \rangle + b_0' + \sum_{j=1}^{d_1'} u_j' \phi \left( \langle \boldsymbol{w}_j', \boldsymbol{x} \rangle + b_j' \right). \tag{2}$$

We will also prove a similar result for the case that the algorithm is allowed to query the network just on points in $\mathbb{R}_+^d$, but on the other hand, equation equation 2 needs to be satisfied just for $\boldsymbol{x} \in \mathbb{R}_+^d$. This case is essential for reconstructing depth-three networks, and we will call it the $\mathbb{R}_+^d$-restricted case.

**Theorem 2.** *In the $\mathbb{R}_+^d$-restricted case there is an algorithm that given an oracle access to a $\delta$-regular network as in equation 1, reconstructs it using $O\left((dd_1 \log(1/\delta) + d_1 d) Q + d^2 d_1^2\right)$ time and $O\left(dd_1 \log(1/\delta) + d_1 d\right)$ queries.*

## 2.2 Depth Three Networks

Consider a 3-layer network given by

$$\mathcal{M}(\boldsymbol{x}) = \langle \boldsymbol{u}, \phi(\boldsymbol{V}\phi(\boldsymbol{W}\boldsymbol{x} + \boldsymbol{b}) + \boldsymbol{c}) \rangle \tag{3}$$

where $\boldsymbol{W} \in \mathbb{R}^{d_1 \times d}$, $\boldsymbol{b} \in \mathbb{R}^{d_1}$, $\boldsymbol{V} \in \mathbb{R}^{d_2 \times d_1}$, $\boldsymbol{c} \in \mathbb{R}^{d_2}$, $\boldsymbol{u} \in \mathbb{R}^{d_2}$ and $\phi$ is the ReLU function defined element-wise. We assume $\boldsymbol{W}, \boldsymbol{V}, \boldsymbol{u}, \boldsymbol{b}, \boldsymbol{c}$, along with $d_1$ and $d_2$, are unknown to the user, which have only black box access to $\mathcal{M}(x)$ for any $\boldsymbol{x} \in \mathbb{R}^d$. Besides $\delta$-regularity we will assume that (i) $d_1 \leq d$ and that (ii) the top layer has non-zero partial derivatives: For the second layer function $F : \mathbb{R}^{d_1} \to \mathbb{R}$ given by $F(\boldsymbol{x}) = \langle \boldsymbol{u}, \phi(V\boldsymbol{x} + \boldsymbol{c}) \rangle$ we assume that for any $\boldsymbol{x} \in \mathbb{R}^{d_1}_+$ and $j \in [d_1]$, the derivative of $F$ in the direction of $\boldsymbol{e}^{(j)}$ and $-\boldsymbol{e}^{(j)}$ is not zero. We note that if $d_2$ is large compared to $d_1$ ($d_2 \geq 3.5d_1$ would be enough) this assumption is valid for most choices of $\boldsymbol{u}, \boldsymbol{V}$ and $\boldsymbol{c}$ (see theorem 5).

**Theorem 3.** *There is an algorithm that given an oracle access to a $\delta$-regular network as in equation 2, with $d_1 \leq d$ and top layer with non-zero partial derivatives, reconstruct it using* $\mathrm{poly}(d, d_1, d_2, \log(1/\delta))$ *time and queries.*

By reconstruction we mean that the algorithm will find $d'_1, d'_2 \in \mathbb{N}$, weights $\boldsymbol{v}'_0, \ldots, \boldsymbol{v}'_{d'_2} \in \mathbb{R}^{d'_1}$, $c'_0, \ldots, c'_{d'_2} \in \mathbb{R}$, $u'_1, \ldots, u'_{d'_2} \in \mathbb{R}$, as well as a matrix $\boldsymbol{W}' \in \mathbb{R}^{d'_1 \times d}$ and a vector $\boldsymbol{b}' \in \mathbb{R}^{d'_1}$ such that

$$\forall \boldsymbol{x} \in \mathbb{R}^d, \ \ \mathcal{M}(\boldsymbol{x}) = \langle \boldsymbol{v}'_0, \phi(\boldsymbol{W}'\boldsymbol{x} + \boldsymbol{b}') \rangle + c'_0 + \sum_{j=1}^{d'_2} u'_j \phi \left( \langle \boldsymbol{v}'_j, \phi(\boldsymbol{W}'\boldsymbol{x} + \boldsymbol{b}') \rangle + c'_j \right).$$

## 2.3 Novelty of the Reconstructions

Having an *exact* reconstruction is an essential task for extracting a model. While approximate reconstructions, such as in Chen et al. (2021), may mimic the output of the extracted network, they cannot reveal information on the architecture, like the network's width. Moreover, an approximated reconstruction can be viewed as a regression task. For example, the work of Shi et al. (2017) used Naive Bayes and SVM models to predict the network's output. An exact reconstruction requires building new tools, as we provide in this work.

Exploring the non-linearity parts of a network can offer information on the relations between the weights of a neuron up to a multiplicative factor. Specifically, the sign of a neuron is missing. Indeed: for the $j$'th neuron both $(\boldsymbol{w}_j, b_j)$ and $(-\boldsymbol{w}_j, -b_j)$ have the property of breaking the linearity of $\mathcal{M}(\boldsymbol{x})$ at the same values of $\boldsymbol{x}$. To achieve the global signs of all the neurons, one requires either to restrict the width of the network (as in Milli et al. (2019)) or to use brute-force over all possible combinations (as in Carlini et al. (2020) and Rolnick & Körding (2020)). We bypass this challenge by allowing reconstruction up to an affine transformation and using the fact that for all $\boldsymbol{x} \in \mathbb{R}^d$,

$$\langle \boldsymbol{w}, \boldsymbol{x} \rangle + b = \phi(\langle \boldsymbol{w}, \boldsymbol{x} \rangle + b) - \phi(-\langle \boldsymbol{w}, \boldsymbol{x} \rangle - b).$$

This bypass allows the reconstruction of a network with any finite width in a polynomial time.

Another technical novelty of the paper is an algorithm that can identify whether a neuron belongs to the first or the second layer. This allows us to handle a second hidden layer after peeling the first layer.

## 3 Proofs

### 3.1 Notations and Terminology

We denote by $\boldsymbol{e}^{(1)}, \ldots, \boldsymbol{e}^{(d)}$ the standard basis of $\mathbb{R}^d$ and by $\mathbb{B}(\boldsymbol{x}, \delta)$ the open ball around $\boldsymbol{x} \in \mathbb{R}^d$ with radius $\delta > 0$. For $\boldsymbol{w} \in \mathbb{R}^d$ and $b \in \mathbb{R}$ we denote by $\Lambda_{\boldsymbol{w}, b}$ the affine function $\Lambda_{\boldsymbol{w}, b}(\boldsymbol{x}) = \langle \boldsymbol{w}, \boldsymbol{x} \rangle + b$. For a point $\boldsymbol{x} \in \mathbb{R}^d$ and a set $A \subset \mathbb{R}^d$ we denote by $d(\boldsymbol{x}, A) = \inf_{\boldsymbol{y} \in A} \|\boldsymbol{x} - \boldsymbol{y}\|$ the distance between $\boldsymbol{x}$ and $A$. Given a subspace $\mathbb{P}$, A *Gaussian in* $\mathbb{P}$ is a Gaussian vector $\mathbf{x}$ in $\mathbb{R}^d$ whose density function is supported in $\mathbb{P}$. We say that it is standard if the projection of $\mathbf{x}$ on any line in $\mathbb{P}$ that passes through $\mathbb{E}[\mathbf{x}]$ has a variance of 1.

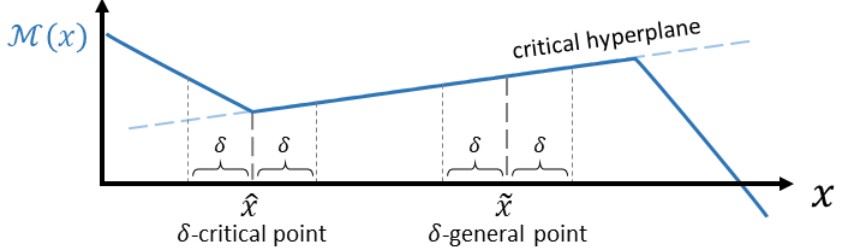

Figure 1: An illustration of one-dimensional piecewise linear function $\mathcal{M} : \mathbb{R} \to \mathbb{R}$

The *state* of a neuron on a point $x \in \mathbb{R}^d$ is the sign of the input of that neuron (either positive, negative, or zero). The *state* of a network on a point $x \in \mathbb{R}^d$ is a description of the states of all neurons at $x$. Similarly, the state of the first layer at $x \in \mathbb{R}^d$ is a description of the state of all first layer neurons at $x$.

The angle between a hyperplane $\mathbb{P}$ with a normal vector $n$ and a line $\{tx + y : t \in \mathbb{R}\}$ (or just a vector $x \neq 0$) is defined as $\left| \left\langle \frac{x}{\|x\|}, n \right\rangle \right|$. Likewise, the *distance* between two hyperplanes $\mathbb{P}_1, \mathbb{P}_2$ with normal vectors $n_1, n_2$ respectively, is given by $D(\mathbb{P}_1, \mathbb{P}_2) := \sqrt{1 - \langle n_1, n_2 \rangle^2}$. We say that a hyperplane is $\delta$-*general* if its angle with all the $d$ axes is at least $\delta$. A hyperplane is *general* if it is $\delta$-general for some $\delta > 0$ (equivalently, it is not parallel to any axis).

## 3.2 PIECEWISE LINEAR FUNCTIONS

Let $f : \mathbb{R}^d \to \mathbb{R}$ be piecewise linear, with finitely many pieces. A *general point* is a point $x \in \mathbb{R}^d$ such that exists a neighborhood around $x$ for which $f$ is affine in that neighborhood. Furthermore, we say that the point $x \in \mathbb{R}^d$ is a $\delta$-*general point* if $f$ is affine in $\mathbb{B}(x, \delta)$. Complementarily, a *critical point* is a point $x \in \mathbb{R}^d$ such that for every $\delta > 0$, $f$ is not affine in $\mathbb{B}(x, \delta)$. A *critical hyperplane* is an affine hyperplane $\mathbb{P}$, whose intersection with the set of critical points is of dimension $d - 1$. For a critical hyperplane $\mathbb{P}$, we say that a point $x \in \mathbb{R}^d$ is $\mathbb{P}$-*critical* if it is critical and $x \in \mathbb{P}$. Figure 3.2 illustrates the above definitions for the one-dimensional input case.

Note that there are finitely many critical hyperplanes for any piecewise linear function, that any critical point belongs to at least one critical hyperplane, and most[1] critical points belong to exactly one critical hyperplane. We will call such points *non-degenerate*. Furthermore, we will say that a critical point $x$ is $\delta$-*non-degenerate* if exactly one critical hyperplane intersects with $\mathbb{B}(x, \delta)$.

For the function $\mathcal{M}$ computed by a network such as equation 1 or equation 3, we note that for any $j \in [d_1]$, the hyperplane $\mathbb{P}_j = \{x : \langle w_j, x \rangle + b_j = 0\}$ is a critical hyperplane. In this case, we say that $\mathbb{P}$ corresponds to the $j$th neuron, and vice-versa. Also, if $x$ is a critical point, then at least one of the neurons is in a critical state (i.e., its input is 0). In this case, we will say that $x$ is a critical point of that neuron.

We next describe a few simple algorithms related to piecewise linear functions that we will use frequently. Their correctness is given in section D of the appendix; here we briefly sketch the idea behind it.

### 3.2.1 RECONSTRUCTION OF AN AFFINE FUNCTION

We note that if $x$ is an $\epsilon$-general point of a function $f$, then one can reconstruct the affine function $f$ computes over $\mathbb{B}(x, \epsilon)$ with $d + 1$ queries in $\mathbb{B}(x, \epsilon)$ and $O(dQ)$ time. Algorithm 2 reconstructs the desired affine function.

---

[1] By most, we mean all except a set whose dimension is $d - 2$.

### 3.2.2 RECONSTRUCTION OF CRITICAL POINTS IN ONE DIMENSION

We say that a piecewise linear one dimensional function $f : \mathbb{R} \to \mathbb{R}$ is $\delta$-*nice* if: (1) All its critical points are in $\left(-\frac{1}{\delta}, \frac{1}{\delta}\right) \setminus (-\delta, \delta)$, (2) each piece is of length at least $\delta$, (3) there are no two pieces that share the same affine function, and (4) all the points in the grid $\frac{2^{-\lceil \log_2(2/\delta^2) \rceil}}{\delta} \mathbb{Z}$ are $\delta^2$-general.

Given a $\delta$-nice function, algorithm 1 recovers the left-most critical point in the range $(a, 1/\delta)$, if such a point exist, using $O\left(\log(1/\delta)Q\right)$ time. In short, the algorithm works similar to a binary search, where each iteration splits the current range into two halves and keeps the left half if and only if it is not affine.

---

**Algorithm 1** FIND_CP$(\delta, f, a)$: Single critical point reconstruction

---

**Input:** Parameter $\delta < 1$, black box access to a $\delta$-nice $f : \mathbb{R} \to \mathbb{R}$, and left limit $a \in \left(-\frac{1}{\delta}, \frac{1}{\delta}\right)$
**Output:** The left most critical point of $f$ in $(a, 1/\delta)$.
 1: Set $x_L = -\frac{1}{\delta}, x_R = \frac{1}{\delta}$
 2: **for** $j = 1, \dots, \lceil \log_2(2/\delta^2) \rceil + 1$ **do**
 3:   If $\frac{x_L + x_R}{2} \leq a$ or $\text{AFFINE}_{\delta^2}\left(f, x_L\right) = \text{AFFINE}_{\delta^2}\left(f, \frac{x_L + x_R}{2}\right)$, set $x_L = \frac{x_L + x_R}{2}$. Else, set $x_R = \frac{x_L + x_R}{2}$.
 4: **end for**
 5: Let $\Lambda_L = \text{AFFINE}_{\delta^2}\left(f, x_L\right)$ and $\Lambda_R = \text{AFFINE}_{\delta^2}\left(f, x_R\right)$. If $\Lambda_L = \Lambda_R$ then return "no critical points in $(a, 1/\delta)$". Else, return the point $x$ for which $\Lambda_L(x) = \Lambda_R(x)$

---

With algorithm 1 we can reconstruct all the critical points of $f$ in a given range $(a, b) \subset (-1/\delta, 1/\delta)$ in $O\left(k \log(1/\delta)Q\right)$ time, where $k$ is the number of critical points in $(a, b)$. Indeed, we can invoke algorithm 1 to find the left-most critical point $x_1$ in $(a, b)$, then the one on its right and so on, until there are no more critical points in $(a, b)$.

### 3.2.3 RECONSTRUCTION OF A CRITICAL HYPERPLANE

Let $f : \mathbb{R}^d \to \mathbb{R}$ be a piecewise linear function. Assume that $\boldsymbol{x}$ is a $\delta$-non-degenerate $\mathbb{P}$-critical point. If $\boldsymbol{x}_1, \boldsymbol{x}_2 \in \mathbb{B}(\boldsymbol{x}, \delta)$ are two points on opposite sides of $\mathbb{P}$, then $\mathbb{P}$ is the null space of $\Lambda_1 - \Lambda_2$, where $\Lambda_1, \Lambda_2$ are the affine functions computed by $f$ near $\boldsymbol{x}_1$ and $\boldsymbol{x}_2$. Algorithm 3 therefore reconstructs $\mathbb{P}$ in $O(dQ)$ time.

### 3.2.4 CHECKING CONVEXITY/CONCAVITY IN A $\delta$-NON-DEGENERATE CRITICAL POINT

Let $f : \mathbb{R}^d \to \mathbb{R}$ be a piecewise linear function. Assume that $\boldsymbol{x}$ is a $\delta$-non-degenerate $\mathbb{P}$-critical point. As $\mathbb{P}$ is the intersection of exactly two affine functions, then $f$ is necessarily convex or concave in $\mathbb{B}(\boldsymbol{x}, \delta)$. Furthermore, for any unit vector $\boldsymbol{e}$ that is not parallel[2] to $\mathbb{P}$, we have that $f$ is convex in $\mathbb{B}(\boldsymbol{x}, \delta)$ if and only if it is convex in $[\boldsymbol{x} - \delta \boldsymbol{e}, \boldsymbol{x} + \delta \boldsymbol{e}]$, in which case the slope of $t \mapsto f(\boldsymbol{x} + t\boldsymbol{e})$ in $[-\delta, 0]$ is strictly smaller then its slope in $[0, \delta]$. Algorithm 4 therefore determine if $f$ is convex or concave in $\mathbb{B}(\boldsymbol{x}, \delta)$ in $O(Q)$ time.

### 3.2.5 DISTINGUISH $\epsilon$-GENERAL POINT FROM $\epsilon$-NON-DEGENERATE CRITICAL POINT

Let $f : \mathbb{R}^d \to \mathbb{R}$ be a piecewise linear function. Assume that $\boldsymbol{x}$ is either a $\epsilon$-non-degenerate $\mathbb{P}$-critical point or an $\epsilon$-general point. Then by the definitions, for any unit vector $\boldsymbol{e}$ that is not parallel to $\mathbb{P}$, $\boldsymbol{x}$ is critical if and only if the slope of $t \mapsto f(\boldsymbol{x} + t\boldsymbol{e})$ is different in the segments $[-\epsilon, 0]$ and $[0, \epsilon]$. Algorithm 5 therefore determine if $\boldsymbol{x}$ is critical in $O(Q)$ time.

## 3.3 GENERAL POSITION ASSUMPTION

We say that a two-layers network as in equation 1 is $\delta$-*regular* if the conditions for the inputs of algorithms 2-5 are met for the network and for any critical point that lies on the standard axes. For a three-layer network, as in equation 3, we also require that the above apply to the sub-network defined by the top two layers. A two- and three-layers network is called *regular* if it is $\delta$-regular

---

[2]By parallel we mean that the vector is orthogonal to the hyperplane's normal.

**Algorithm 2** $\texttt{AFFINE}_\epsilon(f, \boldsymbol{x})$ –
Affine map reconstruction from $\epsilon$-general point

**Input:** Black box access to a piecewise linear $f : \mathbb{R}^d \to \mathbb{R}$, parameter $\epsilon > 0$, and an $\epsilon$-general point $\boldsymbol{x} \in \mathbb{R}^d$
**Output:** Vector $\boldsymbol{w} \in \mathbb{R}^d$ and $b \in \mathbb{R}$ such that $\forall \boldsymbol{y} \in \mathbb{B}(\boldsymbol{x}, \epsilon)$, $\Lambda_{\boldsymbol{w},b}(\boldsymbol{y}) = f(\boldsymbol{y})$

1: Return $w_i = \frac{f(\boldsymbol{x}+\epsilon \boldsymbol{e}^{(i)})-f(\boldsymbol{x})}{\epsilon}$ and
$b = \left( f(\boldsymbol{x}) - \sum_{i=1}^d \frac{f(\boldsymbol{x}+\epsilon \boldsymbol{e}^{(i)})-f(\boldsymbol{x})}{\epsilon} x_i \right)$

**Algorithm 3** $\texttt{FIND\_HP}(f, \delta, \boldsymbol{x})$ –
Reconstruction of a critical hyperplane

**Input:** Black box access to a piecewise linear $f : \mathbb{R}^d \to \mathbb{R}$, a parameter $\delta > 0$, a $\delta$-non-degenerate $\mathbb{P}$-critical point $\boldsymbol{x} \in \mathbb{R}^d$ for $\delta$-general $\mathbb{P}$
**Output:** $\boldsymbol{w} \in \mathbb{R}^d$ and $b \in \mathbb{R}$ such that $\mathbb{P} = \{\boldsymbol{x} : \Lambda_{\boldsymbol{w},b}(\boldsymbol{x}) = 0\}$

1: Set $\epsilon = (\delta/2)^2$
2: Using algorithm 2 obtain $(\boldsymbol{w}_1, b_1) = \texttt{AFFINE}_\epsilon\left(f, \boldsymbol{x} + \frac{\delta}{2}\boldsymbol{e}^{(1)}\right)$ and $(\boldsymbol{w}_2, b_2) = \texttt{AFFINE}_\epsilon\left(f, \boldsymbol{x} - \frac{\delta}{2}\boldsymbol{e}^{(1)}\right)$
3: Return $\boldsymbol{w} = \boldsymbol{w}_1 - \boldsymbol{w}_2$ and $b = b_1 - b_2$

**Algorithm 4** $\texttt{IS\_CONVEX}(f, \delta, \boldsymbol{x})$ –
Checking convexity/concavity

**Input:** Black box access to a piecewise linear $f : \mathbb{R}^d \to \mathbb{R}$, a parameter $\delta > 0$, a $\delta$-non-degenerate $\mathbb{P}$-critical point $\boldsymbol{x} \in \mathbb{R}^d$ for general $\mathbb{P}$
**Output:** Is $\boldsymbol{x}$ convex in $f$ at $\mathbb{B}(\boldsymbol{x}, \delta)$

1: **if** $f(\boldsymbol{x}+\delta \boldsymbol{e}^{(1)})-f(\boldsymbol{x}) > f(\boldsymbol{x})-f(\boldsymbol{x}-\delta \boldsymbol{e}^{(1)})$ **then**
2:     Return "convex"
3: **else**
4:     Return "concave"
5: **end if**

**Algorithm 5** $\texttt{IS\_GENERAL}(f, \epsilon, \boldsymbol{x})$ –
Distinguish general point from critical point

**Input:** Black box access to a piecewise linear $f : \mathbb{R}^d \to \mathbb{R}$, a parameter $\epsilon > 0$, a point $\boldsymbol{x}$ that is either $\epsilon$-general or $\epsilon$-non-degenerate $\mathbb{P}$-critical point for general $\mathbb{P}$
**Output:** Is $\boldsymbol{x}$ general?

1: **if** $f(\boldsymbol{x}+\epsilon \boldsymbol{e}^{(1)})-f(\boldsymbol{x}) = f(\boldsymbol{x})-f(\boldsymbol{x}-\epsilon \boldsymbol{e}^{(1)})$ **then**
2:     Return "general", else return "critical"
3: **end if**

for some $\delta > 0$. A network is in *general position* if it is regular, and for three-layer networks, as in equation 3, we also require $\boldsymbol{W}$ to be surjective and that the top-layer will not have zero partial derivatives. A formal definition for a $\delta$-regular network is given in section A of the appendix. Here we want to state sufficient conditions that ensure the regularity and general position of a network. The proofs are given in section A of the appendix.

**Lemma 1.** *The set of non-regular neural networks as in equation 1 and equation 3 have a zero Lebesgue measure.*

**Lemma 2.** *Let $\mathcal{M}$ be a neural network as in equation 1 or equation 3. Let $q$ be the number of neurons in the network, and let $M > 0$ be an upper bound on the absolute value of the weights. For each weight in the network, add a uniform element in $[-2^{-d}, 2^{-d}]$. Then, the noisy network $\mathcal{M}'$ is $\delta$-regular for $\delta > 0$ such that $\log(1/\delta) = \text{poly}(d \log(qM))$ with probability of $1 - 2^{-d}$.*

**Lemma 3.** *For a general three-layers network as in equation 3, if $d_1 \leq d$ then $\boldsymbol{W}$ is surjective with probability 1.*

**Lemma 4.** *For a general three-layers network as in equation 3, if $3.5d_1 \leq d_2$ then the top layer has non-zero partial derivatives with probability $1 - o(1)$.*

We note that the assumptions in section A may seem lengthy. The keen reader may notice overlaps between some of them and might suggest approaches to avoid others, for example, by adding randomization to the queries. Yet, we keep them as is for the fluency of reading, to emphasize the main concepts of the extraction. As training a network in practice begins from a random initialization, it is very likely for the network to be found in a regular position after the learning phase. Therefore, we took the freedom to ignore unlikely positions instead of combining them under a very restrictive rule.

## 3.4 Reconstruction of Depth Two Network – Sketch Proof of Theorems 1 and 2

Recall that our goal is to recover a depth-two network in the form of equation equation 1. We will assume without loss of generality that the $u_i$'s are in $\{\pm 1\}$, as any neuron $\boldsymbol{x} \mapsto u\phi(\langle \boldsymbol{w}, \boldsymbol{x} \rangle + b)$ calculates the same function as $\boldsymbol{x} \mapsto \frac{u}{|u|}\phi(\langle |u|\boldsymbol{w}, \boldsymbol{x} \rangle + |u|b)$, as ReLU is a positive homogeneous function.

Our algorithm will first find a critical point for each neuron. For a regular network, each critical hyperplane intersects the axis $\mathbb{R}e^{(1)}$ exactly once, so we can reconstruct such a set of critical points by invoking algorithm 1 on the function $t \mapsto \mathcal{M}(te^{(1)})$.

We next reconstruct a single neuron corresponding to a given critical point $\boldsymbol{x}$. For simplicity, assume that $\boldsymbol{x}$ is a $\delta$-critical point of the $j$'th neuron. Using algorithm 3 we find an affine function $\Lambda$ such that $\Lambda = \Lambda_{\boldsymbol{w}_j, b_j}$ or $\Lambda = -\Lambda_{\boldsymbol{w}_j, b_j}$. Then, to recover $u_j$, note that if $u_j = 1$ then $\mathcal{M}(x)$ is strictly convex in $\mathbb{B}(\boldsymbol{x}, \delta)$ as the function $u_j\phi(\langle \boldsymbol{w}_j, \boldsymbol{x} \rangle + b_j)$ is convex. Similarly, if $u_j = -1$ then $\mathcal{M}(\boldsymbol{x})$ is strictly concave in $\mathbb{B}(\boldsymbol{x}, \delta)$. Thus, we recover $u_j$ using using algorithm 4.

Finally, note that $\phi(\Lambda(\boldsymbol{x}))$ is either $\phi(\langle \boldsymbol{w}_j, \boldsymbol{x} \rangle + b_j)$ or $\phi(\langle \boldsymbol{w}_j, \boldsymbol{x} \rangle + b_j) - \langle \boldsymbol{w}_j, \boldsymbol{x} \rangle - b_j$. Hence, $u_j\phi(\Lambda(\boldsymbol{x}))$ equals to $u_j\phi(\Lambda_{\boldsymbol{w}_j, b_j}(\boldsymbol{x}))$ up to an affine map. The approach is detailed in Algorithm 6.

---

**Algorithm 6** Recover depth-two network

---

**Input:** Parameter $\delta$ and a black box access to a $\delta$-regular network $\mathcal{M}$ as in equation 1
**Output:** Weights such that for all $\boldsymbol{x}$, $\mathcal{M}(\boldsymbol{x}) = \Lambda_{\boldsymbol{w}_0', b_0'}(\boldsymbol{x}) + \sum_{i=1}^{m} u_i'\phi\left(\Lambda_{\boldsymbol{w}_i', b_i'}(\boldsymbol{x})\right)$

1: Use repeatedly $\text{FIND\_CP}(\delta, t \mapsto \mathcal{M}(te^{(1)}), \cdot)$ to find all the critical points on the axis $\{te^{(1)} : t \in \mathbb{R}\}$ (see section 3.2). Denote these points by $\boldsymbol{x}_1, \ldots, \boldsymbol{x}_m$.
2: **for** $i = 1, \ldots, m$ **do**
3:     Compute $(\boldsymbol{w}_i', b_i') = \text{FIND\_HP}(\mathcal{M}, \delta, \boldsymbol{x}_i)$.
4:     If $\text{IS\_CONVEX}(\mathcal{M}, \delta, \boldsymbol{x}_i) = $ "convex" then set $u_i' = 1$. Else, set $u_i' = -1$.
5: **end for**
6: Calc $(\boldsymbol{w}_0', b_0') = \text{AFFINE}_\delta\left(\boldsymbol{x} \mapsto \mathcal{M}(\boldsymbol{x}) - \sum_{i=1}^{m} u_i'\phi\left(\Lambda_{\boldsymbol{w}_i', b_i'}(\boldsymbol{x})\right), \text{x}'\right)$ for a random $\text{x}' \in \mathbb{R}^d$.
7: Return the function $\boldsymbol{x} \mapsto \Lambda_{\boldsymbol{w}_0', b_0'}(\boldsymbol{x}) + \sum_{i=1}^{m} u_i'\phi\left(\Lambda_{\boldsymbol{w}_i', b_i'}(\boldsymbol{x})\right)$.

---

The following theorem proves the correctness of algorithm 6, and implies theorem 1. The proof is given in section B of the appendix.

**Theorem 4.** *Algorithm 6 reconstruct a $\delta$-regular network in time* $O\left((\log(1/\delta) + d)d_1 Q + d^2 d_1\right)$.

### 3.4.1 SKETCH PROOF OF THEOREM 2

Our algorithm for reconstruction of depth-two networks can be easily modified to work in the $\mathbb{R}_+^d$-restricted setting, with the difference that in order to reconstruct a $\delta$-critical point for each neuron (step 1 in algorithm 6), we will need to search in the range $\left(0, \frac{1}{\delta}\right)e^{(i)}$ for all $i \in [d]$, as a critical hyperplane of a given neuron might not intersect with $\mathbb{R}_+ e^{(1)}$. Because of this change, each neuron might be discovered several times (up to $d$ times), and we will need an additional step that combines neurons with the same affine map (up to a sign). For the particular case where the neuron has no critical points on the positive orthant, one can ignore it without affecting equation equation 2 for all $\boldsymbol{x} \in \mathbb{R}_+^d$.

These changes will result in a total runtime of $O(dd_1 \log(1/\delta)Q)$ instead of $O(d_1 \log(1/\delta)Q)$ for step 1, $O(d^2 d_1 Q)$ instead of $O(dd_1 Q)$ for the loop, and $O(d_1^2 d^2)$ for combining similar neurons. The total runtime will therefore be $O((\log(1/\delta) + d)dd_1 Q + d^2 d_1^2)$. A formal proof is given in section B of the appendix.

### 3.5 RECONSTRUCTION OF DEPTH THREE NETWORK – SKETCH PROOF OF THEOREM 3

Recall that our goal is to recover a $\delta$-regular network of the form

$$\mathcal{M}(\boldsymbol{x}) = \langle \boldsymbol{u}, \phi(\boldsymbol{V}\phi(\boldsymbol{W}\boldsymbol{x} + \boldsymbol{b}) + \boldsymbol{c}) \rangle.$$

We denote by $\boldsymbol{w}_j$ the $j$th row of $\boldsymbol{W}$ and assume without loss of generality that it is of unit norm, as any neuron of the form $\boldsymbol{x} \mapsto \phi(\langle \boldsymbol{w}, \boldsymbol{x} \rangle + b)$ can be replaced by $\boldsymbol{x} \mapsto \|\boldsymbol{w}\|\phi\left(\left\langle \frac{\boldsymbol{w}}{\|\boldsymbol{w}\|}, \boldsymbol{x} \right\rangle + \frac{b}{\|\boldsymbol{w}\|}\right)$. Likewise, and similar to our algorithm for reconstruction of depth-two networks, we will assume that $\boldsymbol{u} \in \{\pm 1\}^{d_2}$.

The algorithm will be decomposed into four steps described in the following four subsections. In the first step, we will extract a set of critical hyperplanes that contains all the critical hyperplanes

that correspond to a first layer neuron. In the second step, we will prune this list and will be left with a list that contains precisely the critical hyperplanes that correspond to a first layer neuron. In the third step, we will use this list to recover the first layer. Once the first layer is recovered, as the fourth step, we recover the second layer via a reduction to the problem of recovering a depth-two network.

### 3.5.1 Extracting a set containing the critical hyperplanes of the first layer

For the first step, we find a list $L = \left\{ (\boldsymbol{x}_1, \hat{\mathbb{P}}_1), \ldots, (\boldsymbol{x}_m, \hat{\mathbb{P}}_m) \right\}$ of pairs such that:

- For each $k$, $\hat{\mathbb{P}}_k$ is a critical hyperplane of $\mathcal{M}$ and $\boldsymbol{x}_k$ is a $\delta$-non-degenerate critical point whose critical hyperplane is $\mathbb{P}_k$
- The list contains all the critical hyperplanes of first-layer neurons

We find those points using Algorithm 3. Note that $m = O(d_1 d_2)$ (e.g. Telgarsky (2016)). Let $\hat{\mathbb{P}}_1, \ldots, \hat{\mathbb{P}}_m$ be the critical hyperplanes corresponding to these points, found using Algorithm 3. Finally, lemma 8 below, together with $\delta$-regularity implies that every hyperplane $\mathbb{P}$ that corresponds to a first layer neuron intersects $\mathbb{R}\boldsymbol{e}^{(1)}$ exactly once, and this intersection point is a $\delta$-non-degenerate $\mathbb{P}$-critical point.

### 3.5.2 Identifying first layer critical hyperplanes

The next step is to take the list $L = \left\{ (\boldsymbol{x}_1, \hat{\mathbb{P}}_1), \ldots, (\boldsymbol{x}_m, \hat{\mathbb{P}}_m) \right\}$ from the previous step, verify all the planes corresponding to first-layer neurons and remove all the other hyperplanes. The idea behind this verification is simple: If $\mathbb{P}$ corresponds to a neuron in the first layer then any point in $\mathbb{P}$ is a critical point of $\mathcal{M}$ (see lemma 8). On the other hand, if $\mathbb{P}$ corresponds to a neuron in the second layer, then not all its points are critical for $\mathcal{M}$. Moreover, intersections with hyperplanes from the first layer change the input for the second layer neurons, hence creating a new piece that replaces $\mathbb{P}$. Thus, in order to verify if $\mathbb{P}$ corresponds to a first layer neuron, we will go over all the hyperplanes $\hat{\mathbb{P}}_k \in L$, and for each of them, will find a point $\boldsymbol{x}' \in \mathbb{P}$ that is the opposite side of $\hat{\mathbb{P}}_k$ (relative to $\boldsymbol{x}$) and will check if it is critical. If it is not critical for one of the hyperplanes, we know that $\mathbb{P}$ does not correspond to a first layer neuron. If all the points that we have examined are critical, even for $\hat{\mathbb{P}}_k$ corresponded to a first layer neuron, then $\boldsymbol{x}'$ is critical, which means that $\mathbb{P}$ must correspond to a first layer neuron.

Algorithm 7 implements this idea. There is one caveat that we need to handle: The examined point has to be generic enough in order to test whether it is critical or not using algorithm 5. To make sure that the point is general enough, we slightly perturb it. The correctness of the algorithm follows from lemmas 10 and 11 below. Due to the perturbations, the algorithm has a success probability of at least $1 - \frac{2^{-d}}{m}$ over the choice of $\boldsymbol{x}'$ for each hyperplane and at least $1 - 2^{-d}$ for all hyperplanes. Each step in the for-loop takes $O(dQ)$ operations. As the list size is $O(d_1 d_2)$, the total running time over all hyperplanes is $O(d_1^2 d_2^2 dQ)$.

### 3.5.3 Identifying directions

Since the rows in $\boldsymbol{W}$ are assumed to have a unit norm, the list of the critical hyperplanes of the first-layer neurons, obtained in the previous step, determines the weights up to sign. In order to recover the correct sign of $(\hat{\boldsymbol{w}}_1, \hat{b}_1)$, we can simply do the following test: Choose a point $\boldsymbol{x}$ such that $\hat{\boldsymbol{w}}_1 \boldsymbol{x} + \hat{b}_1 = 0$, and query the network in the points $\boldsymbol{x} + \epsilon \boldsymbol{z}, \boldsymbol{x} - \epsilon \boldsymbol{z}$, for small $\epsilon$, where $\boldsymbol{z} \in \mathbb{R}^d$ is a unit vector that has the property that is orthogonal to $\hat{\boldsymbol{w}}_2, \ldots, \hat{\boldsymbol{w}}_{d_1}$, but $\hat{\boldsymbol{w}}_1 \boldsymbol{z} > 0$. If we assume that $W$ is right invertible, then such a $\boldsymbol{z}$ exists, as $\boldsymbol{w}_1, \ldots, \boldsymbol{w}_{d_1}$ are linearly independent.

Let $out(\boldsymbol{x})$ be the output of the first layer given some point $\boldsymbol{x}$, then:

$$out(\boldsymbol{x} + \epsilon \boldsymbol{z}) = \begin{pmatrix} \langle \boldsymbol{w}_1, \boldsymbol{x} + \epsilon \boldsymbol{z} \rangle + b_1 \\ \langle \boldsymbol{w}_2, \boldsymbol{x} + \epsilon \boldsymbol{z} \rangle + b_2 \\ \vdots \\ \langle \boldsymbol{w}_{d_1}, \boldsymbol{x} + \epsilon \boldsymbol{z} \rangle + b_{d_1} \end{pmatrix} = \begin{pmatrix} \langle \boldsymbol{w}_1, \boldsymbol{x} \rangle & + \langle \boldsymbol{w}_1, \epsilon \boldsymbol{z} \rangle & + b_1 \\ \langle \boldsymbol{w}_2, \boldsymbol{x} \rangle & & + b_2 \\ \vdots & & \\ \langle \boldsymbol{w}_{d_1}, \boldsymbol{x} \rangle & & + b_{d_1} \end{pmatrix} = out(\boldsymbol{x}) + \epsilon \langle \boldsymbol{w}_1, \boldsymbol{z} \rangle \boldsymbol{e}^{(1)}$$

---

**Algorithm 7** Identifying whether a critical hyperplane corresponds to the first layer

---

**Input:** A Black box access to a $\delta$-regular network $\mathcal{M}$ as in equation 3, a list $L = \left\{ (\boldsymbol{x}_1, \hat{\mathbb{P}}_1), \ldots, (\boldsymbol{x}_m, \hat{\mathbb{P}}_m) \right\}$ of pairs as described in section 3.5.1 and a pair $(\boldsymbol{x}, \mathbb{P}) \in L$

**Output:** Does $\mathbb{P}$ correspond to a first layer neuron?

1: Choose $\delta'$ small enough such that $2^{2(d_1+d_2)} \frac{\delta' \sqrt{2}}{\delta \sqrt{\pi}} \leq \frac{2^{-d-1}}{m^2}$

2: Choose $R > 0$ large enough such that $e^{-\frac{(R-\delta')^2}{2}} \leq \frac{2^{-d-1}}{m^2}$

3: **for** any $k \in [m]$, such that $\hat{\mathbb{P}}_k \neq \mathbb{P}$ **do**

4:     Choose a point $\boldsymbol{z} \in \mathbb{P}$ such that $\boldsymbol{z}$ and $\boldsymbol{x}$ are separated by $\hat{\mathbb{P}}_k$, and $d(\boldsymbol{z}, \mathbb{P}) > R$

5:     Choose a standard Gaussian $Z$ in $\mathbb{P}$ whose mean is $\boldsymbol{z}$

6:     If IS_GENERAL($\mathcal{M}, \delta', Z$), return "$\mathbb{P}$ is **not** a first-layer critical hyperplane"

7: **end for**

8: Return "$\mathbb{P}$ is a first-layer critical hyperplane"

---

Therefore, when moving from $\boldsymbol{x}$ to either $\boldsymbol{x} + \epsilon \boldsymbol{z}$ or $\boldsymbol{x} - \epsilon \boldsymbol{z}$, only the first neuron changes, and after the ReLU activation function, only the positive direction will return a different value. Hence, in order to have the correct sign we can do the following: If $\mathcal{M}(\boldsymbol{x}) \neq \mathcal{M}(\boldsymbol{x} + \epsilon \boldsymbol{z})$ then keep $(\hat{\boldsymbol{w}}_1, \hat{b}_1)$. Else, replace it with $(-\hat{\boldsymbol{w}}_1, -\hat{b}_1)$. We repeat this method for all the neurons $j \in [d_1]$.

The above method fails in the special case where both $\boldsymbol{V}\phi(out(\boldsymbol{x} + \epsilon \boldsymbol{z})) + \boldsymbol{c} \leq 0$ and $\boldsymbol{V}\phi(out(\boldsymbol{x} - \epsilon \boldsymbol{z})) + \boldsymbol{c} \leq 0$, which occur if the partial derivatives of the top-layer are zero at $\phi(\boldsymbol{x} + \epsilon \boldsymbol{z})$ and $\phi(\boldsymbol{x} - \epsilon \boldsymbol{z})$. As we showed on section 3.3, this is not expected if the second layer is wide enough.

The runtime of this step is $O(d_1^3 d + d_1 Q)$, as to find $\boldsymbol{z}$ we need to do Gram-Schmidt, which takes $O(d_1^2 d)$, and additional two queries to find the sign.

### 3.5.4 RECONSTRUCTION OF THE TOP TWO LAYERS

After having the weights of the first layer at hand, and since $\boldsymbol{W}$ is assumed to be right invertible, we can directly access the sub-network defined by the top two layers. Namely, given $\boldsymbol{x} \in \mathbb{R}_+^{d_1}$, we can find $\boldsymbol{z} \in \mathbb{R}^d$ such that

$$\boldsymbol{x} = (\phi(\boldsymbol{w}_1 \boldsymbol{z} + b_1), \ldots, \phi(\boldsymbol{w}_{d_1} \boldsymbol{z} + b_{d_1}))$$

e.g., by taking $\boldsymbol{z} = \boldsymbol{W}^{-1}(\boldsymbol{x} - \boldsymbol{b})$ where $\boldsymbol{W}^{-1}$ is a right inverse of $\boldsymbol{W}$. Now, $\mathcal{M}(\boldsymbol{z})$ is precisely the value of the top layer on the input $\boldsymbol{x}$, and the problem boils down to the problem of reconstructing a depth two network in the $\mathbb{R}_+^d$-restricted case, which we already solved.

Now, the cost of a query to the second layer is $Q$ plus the cost of computing $\boldsymbol{z}$, which is $O(dd_1)$. There is also an asymptotically negligible cost of $O(d_1^2 d)$ for computing $\boldsymbol{W}^{-1}$. The runtime of this step is therefore $O((\log(1/\delta) + d_1) d_1 d_2 (Q + dd_1) + d_1^2 d_2^2)$.

## 4 DISCUSSION AND SOCIAL IMPACT

This work continues a set of empirical and theoretical results, showing that extracting a ReLU network given membership queries is possible. Here we prove that two- and three-layer model extraction can be done in polynomial time. Our nonrestrictive assumptions make it feasible to construct a fully connected network, a convolutional network, and many other architectures.

For practical use, our approach suffers several limitations. First, two- and three-layer networks are too shallow in practice. Second, exact access to the black-box network may not be feasible in practice. As the number of output bits is bounded, numerical inaccuracies may affect the reconstruction, especially when $\delta$ is very small. In that regard, our work is mostly theoretical in nature, showing that reconstruction is provably achievable.

Yet, this work raises practical social concerns regarding the potential risks of membership-queries attacks. Extracting the exact parameters and architecture will allow attackers to reveal proprietary information and even construct adversarial examples. Therefore, uncovering those risks and creating a conversation on ways to protect against them is essential.

As empirical evidence shows, we believe it is possible to prove similar results with even fewer assumptions for deeper models and more complex architectures. Furthermore, it might be interesting to investigate the methods of this paper when we restrict the queries and the outputs up to machine precision. We leave those challenges for future works.

ACKNOWLEDGMENTS

This research is supported by ISF grant 2258/19, and ERC grant 101041711

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

## A  REGULAR NETWORKS

**Definition 1.** *A neural network is called $\delta$-regular if it satisfies the following requirements:*

1. *For each $i \in [d]$, the piecewise linear function $t \mapsto \mathcal{M}(te^{(i)})$ is $\delta$-nice as defined in section 3.2.2.*

2. *Any critical point in the axes $\{te^{(i)} : t \in \mathbb{R}\}$ is $\delta$-non-degenerate.*

3. *Each critical hyperplane corresponds to a single neuron.*

4. *The distance between each pair of critical hyperplanes is at least $\delta$*

5. *The angle between any critical hyperplane and an axis is at least $\delta$. I.e., all critical hyperplanes are $\delta$-general.*

6. *Each critical hyperplane $\mathbb{P}$ corresponding[3] to a second layer neuron also corresponds to a single first layer state. That is, the state of the first layer is the same for any $\mathbb{P}$-critical point.*

7. *In the case of depth-three networks, we assume that the above conditions also apply to the sub-network defined by the top two layers.*

While the definition above is lengthy, most of the requirements overlap, and we detailed them separately for ease of analysis. The following lemma shows that a regular network is expected from a random network. As an untrained network begins from a random initialization, it is very likely to be found in some random position after the learning phase. However, we note that some post-processing methods, like weights-pruning, may affect the general position assumption; such cases should be given specific care and are not in the scope of this paper.

**Lemma 5.** *Let $\mathbb{S}$ be the set of networks as in equation 1 and equation 3 that violate at least one of the above:*

1. *For each $i \in [d]$, the piecewise linear function $t \mapsto \mathcal{M}(te^{(i)})$ is nice.*

2. *Any critical point in the axes $\{te^{(i)} : t \in \mathbb{R}\}$ is non-degenerate.*

3. *Each critical hyperplane corresponds to a single neuron.*

4. *The distance between each pair of critical hyperplanes is non-zero.*

5. *All critical hyperplanes are general.*

6. *Each non-zero critical hyperplane $\mathbb{P}$ corresponding to a second layer neuron also corresponds to a single first layer state.*

7. *In the case of depth-three networks, we assume that the above conditions also apply to the sub-network defined by the top two layers.*

*Then $\mathbb{S}$ has a zero Lebesgue measure.*

*Proof.* It is enough to show that each of the above has a zero measure, as a finite sum of sets with zero measure has a zero measure.

The demand of a specific point $x \in \mathbb{R}^d$ to be critical requires some critical hyperplane $\mathbb{P}$ such that $x \in \mathbb{P}$. This imposes a linear constraint on the set of all such possible hyperplanes and reduces their degree of freedom. Any subspace with dimension $< d$ has a zero Lebesgue measure in $\mathbb{R}^d$, which

---

[3]Remember that we assign a (non-degenerate) critical point to a neuron if the value at that neuron, before the ReLU function, is 0. A point $x$ corresponds to the $i$th neuron in the first layer if $\langle w_i, x \rangle + b_i = 0$, and corresponds to the $j$th neuron in the second layer if $\langle v_j, \phi(Wx + b) \rangle + c_j = 0$. A critical hyperplane corresponds to some neuron if there is a non-empty open set $\mathbb{S} \subseteq \mathbb{P}$ where each $x \in S$ corresponds to that neuron.

is also the case of all the possible hyperplanes containing $\boldsymbol{x}$. As a corollary, the set of hyperplanes that contains the points of $\frac{2^{-\lceil \log_2(2/\delta^2)\rceil}}{\delta}\mathbb{Z}$ has also a zero measure, as $\mathbb{Z}$ is sparse in $\mathbb{R}$.

As another corollary, once fixing a plane $\mathbb{P}$, the set of planes collides with $\mathbb{P}$ exactly on the $i$'th axis, $i \in [d]$, is of zero measure as well, which is the case where a critical point on one of the axes to be degenerate. Even a more degenerate case is where two neurons have the same hyperplane, which means both neurons have exactly the same parameters up to a factor. Obviously, this case has a zero measure in $\mathbb{R}^d$, which implies that with probability 1, a finite set of hyperplanes have a non-zero distance between each other. If we consider the points on the $i$'th axis, $\{\boldsymbol{x} \in \mathbb{R}^d : \langle \boldsymbol{x}, \boldsymbol{e}^{(i)}\rangle = 0\}$, as a hyperplane itself, then it is easy to see that a non-general hyperplane also has a zero measure.

For a one-dimensional function to be nice, one must require that no two pieces share the same affine function. For functions of $t \mapsto \mathcal{M}(t\boldsymbol{e}^{(i)})$, there are no two neurons whose $i$'th parameters are the same. Indeed, the opposite case, where two neurons share the exact same parameters, has a zero measure in $\mathbb{R}$.

As for depth-three networks, all the above is valid for the sub-network defined by the top layer. Moreover, we can consider the first-layer state as an affine transformation for the second layer's neurons. Therefore, in order for a second-layer critical hyperplane $\mathbb{P}$ to span two first-layer states, there must be two second-layer neurons that have the same parameters up to an affine transformation whose uniquely defined by the parameters of the first layer. As the set of all those affine transformations is finite, this imposes a finite set of possible constraints, and each has a zero measure. $\qquad\square$

The following two lemmas state the effect of a small perturbation over $\delta$.

**Lemma 6.** *Let $\mathcal{M}$ be a two-layers neural network as in equation 1. Let q be the number of neurons in the network, and let M be an upper bound on the absolute value of the weights. For each weight in the network, add a uniform element in $[-2^{-d}, 2^{-d}]$, and denote the the noisy network by $\mathcal{M}'$. Then:*

1. *For each $i \in [d]$, all critical points of the piecewise linear function $t \mapsto \mathcal{M}(t\boldsymbol{e}^{(i)})$ are in $\left(-\frac{1}{\delta}, \frac{1}{\delta}\right) \setminus (-\delta, \delta)$ with probability $1 - p_1 = 1 - dq\delta(M+1)2^{d+1}$.*

2. *For each $i \in [d]$, each piece in the piecewise linear function $t \mapsto \mathcal{M}(t\boldsymbol{e}^{(i)})$ is of length at least $\delta$ with probability $1 - p_2 = 1 - d2^d q^2 \delta(M+1)$.*

3. *For each $i \in [d]$, all the points in the grid $\frac{2^{-\lceil \log_2(2/\delta^2)\rceil}}{\delta}\mathbb{Z}$ of the piecewise linear function $t \mapsto \mathcal{M}(t\boldsymbol{e}^{(i)})$ are $\delta^2$-general with probability $1 - p_3 = 1 - 3dq\delta$.*

4. *Any critical point in the axes $\{t\boldsymbol{e}^{(i)} : t \in \mathbb{R}\}$ is $\delta$-non-degenerate with probability $1 - p_4 = 1 - d^{3/2}q^2\delta(M+1)2^{d-1}$.*

5. *The distance between each pair of critical hyperplanes is at least $\delta$ with probability $1 - p_5 = 1 - \delta q^2 \sqrt{d}d(M+1)^2 2^{d+1}$.*

6. *The angle between any critical hyperplane and any axis is at least $\delta$ with probability $1 - p_6 = 1 - dq\delta 2^d$.*

*Proof.* Let the $i$th axis to be $\{t\boldsymbol{e}^{(i)} : t \in \mathbb{R}\}$. Denote by $w_{j,i}$ as the $i$th element of $\boldsymbol{w}_j$ and by $\mathrm{w}'_{j,i} = w_{j,i} + \mathrm{r}_{j,i}$ to be the noisy value of $w_{j,i}$, where $\mathrm{r}_{j,i} \sim U([-2^{-d}, 2^{-d}])$. Similarly, let $\mathrm{b}'_j = b_j + \mathrm{s}_j$ to be the noisy value of $b_j$, where $\mathrm{s}_j \sim U([-2^{-d}, 2^{-d}])$. Then the $j$th neuron has a critical point on the $i$th axis when $t\boldsymbol{e}^{(i)} = \mathrm{t}_{j,i}\boldsymbol{e}^{(i)}$ where $\mathrm{t}_{j,i} = -\frac{\mathrm{b}'_j}{\mathrm{w}'_{j,i}}$. Note that from lemma 5, we have almost surely that $-\infty < \mathrm{t}_{j,i} < \infty$.

1. For each $i \in [d]$ and $j \in [q]$, note that $\left|\mathrm{w}'_{j,i}\right| \leq M + 2^{-d} \leq M + 1$. Given $\alpha \in (0, 2^{-d})$, we have that:
$$p\left(\left|\mathrm{w}'_{j,i}\right| > \alpha\right) \geq p\left(\left|\mathrm{r}_{j,i}\right| > \alpha\right) = 1 - \frac{\alpha}{2^{-d}}.$$

Now, with probability $1 - \alpha 2^{d+1}$ we have that both $\left|w'_{j,i}\right| > \alpha$ and $\left|b'_j\right| > \alpha$, and therefore, by setting $\alpha = \delta(M+1)$:

$$\delta = \frac{\alpha}{M+1} < |t_{j,i}| = \left|\frac{b'_j}{w'_{j,i}}\right| < \frac{M+1}{\alpha} = \frac{1}{\delta}.$$

To make the above valid for every $i \in [d]$ and $j \in [q]$, we can use the union bound to get an overall probability $1 - p_1$ where:

$$p_1 \leq dq\alpha 2^{d+1} = dq\delta(M+1)2^{d+1}.$$

2. Assume the weights were perturbed in the following order: First, $\mathbf{W}'$ is perturbed. Second, the bias of the first neuron, $b_1$, is defined, which sets its critical points with the axes, $t_{1,1}, \ldots, t_{1,d}$. As for the second neuron, we can ask what is the probability for $b_2$ to have a critical point that is $\delta$-close to a critical point of the first neuron. That is, for some $i \in [d]$,

$$p(t_{2,i} \in B(t_{1,i}, \delta)) = p\left(-\frac{b_2 + r_2}{w'_{2,i}} \in (t_{1,i} - \delta, t_{1,i} + \delta)\right)$$
$$= p\left(r_2 \in \left(-w'_{2,i}(t_{1,i} + \delta) - b_2, -w'_{2,i}(t_{1,i} - \delta) - b_2\right)\right)$$
$$\leq 2^d \delta w'_{2,i} \leq 2^d \delta(M+1)$$

and using the union bound,

$$p\left(\exists i \in [d], \, t_{2,i} \in B(t_{1,i}, \delta)\right) \leq d2^d \delta(M+1).$$

Now, let us continue with the perturbation, and for the $j$th neuron, note that the probability to intersect with any of the balls with radius $\delta$ around $t_{1,i}, \ldots, t_{j-1,i}$, $i \in [d]$, is at most $(j-1)d2^d \delta(M+1)$.

Finally, the probability that all the pieces for all $i \in [d]$ are of length at least $\delta$ is $1 - p_2$ with:

$$p_2 \leq \sum_{j=2}^{q}(j-1)d2^d \delta(M+1) \leq d2^d q^2 \delta(M+1).$$

3. Fix $\mathbf{W}'$ and some $i \in [d]$. Note that for the $j$th neuron, $t_{j,i}$ is uniform in $L = \left[\frac{-b_j - 2^{-d}}{w_{j,i}}, \frac{-b_j + 2^{-d}}{w_{j,i}}\right]$. As $L$ is bounded, it intersects with the grid at most $k = \frac{|L|\delta}{2^{-\lceil \log_2(2/\delta^2)\rceil}}$ times. Therefore, for all the points in the grid to be $\delta^2$-general, it means that a segment of length $2\delta^2 k$ should not contain a critical point. As $t_{j,i}$ is uniform, the probability of avoiding that segment is therefore:

$$1 - \frac{2\delta^2 k}{|L|} = 1 - \delta^3 2^{\lceil \log_2(2/\delta^2)\rceil} \geq 1 - 3\delta$$

where the last inequality follows for $\delta \leq 1$.

Overall, we get that the points in the grid are $\delta^2$-general with probability $1 - p_3$, where

$$p_3 = 3dq\delta.$$

4. Let $\mathbb{P}_j$ the critical hyperplane defined by the $j$th neuron. The distance between $\mathbb{P}_j$ and a critical point $t_{k,i}e^{(i)}$, $k \neq j$ is

$$D(\mathbb{P}_j, t_{k,i}) = \frac{\left|\langle \mathbf{w}'_j, t_{k,i}e^{(i)}\rangle + b'_j\right|}{\|\mathbf{w}_j\|} \geq \frac{\left|t_{k,i}w'_{j,i} + b_j + s_j\right|}{\sqrt{d}(M+1)}.$$

As $s_j$ is a symmetric distribution around 0, we have with probability $\geq \frac{1}{2}$ that $\left|t_{k,i}w'_{j,i} + b_j + s_j\right| \geq |s_j|$ and with probability $\frac{1}{2} - \alpha 2^{d-1}$ we have that

$\left|\mathrm{t}_{k,i}\mathrm{w}'_{j,i} + b_j + \mathrm{s}_j\right| \geq |\mathrm{s}_j| \geq \alpha$. If we set $\alpha = \delta\sqrt{d}(M+1)$ then using the union bound we get:

$$p\left(\exists i \in [d], j \neq k, \text{ s.t. } D(\mathbb{P}_j, \mathrm{t}_{k,i}) \leq \delta\right) \leq dq^2\left\{\delta\sqrt{d}(M+1)2^{d-1} - \frac{1}{2}\right\}$$

$$\leq d^{3/2}q^2\delta(M+1)2^{d-1} = p_4.$$

Note that if $\mathrm{t}_{k,i}\boldsymbol{e}^{(i)}$ is far from every other critical hyperplane with at least $\delta$, then it is $\delta$-non-degenerate. Therefore, all the critical points on the axes are $\delta$-non-degenerate with probability $1 - p_4$.

5. For any unit vector $\boldsymbol{e}$ we have that at least one of the coordinates is of absolute value at least $1/\sqrt{d}$. Thus, $p\left(\langle\mathbf{w}', \boldsymbol{e}\rangle \in \delta'\left[-\frac{2^{-d}}{\sqrt{d}}, \frac{2^{-d}}{\sqrt{d}}\right]\right) \leq \delta'$ and $\langle\boldsymbol{w}, \boldsymbol{e}\rangle^2 \leq \|\boldsymbol{w}\|^2 - \langle\boldsymbol{w}, \boldsymbol{e}'\rangle^2 \leq \|\boldsymbol{w}\|^2 - \frac{\delta'2^{-d}}{\sqrt{d}}$ w.p. at least $1 - \delta'$. It follows that $\frac{\langle\boldsymbol{w},\boldsymbol{e}\rangle^2}{\|\boldsymbol{w}\|^2} \leq 1 - \frac{\delta'2^{-d}}{\sqrt{d}\|\boldsymbol{w}\|^2} \leq 1 - \frac{\delta'2^{-d}}{\sqrt{d}d(M+1)^2}$ w.p. at least $1 - \delta'$. Taking roots we get that $\left\langle\frac{\boldsymbol{w}}{\|\boldsymbol{w}\|}, \boldsymbol{e}\right\rangle \leq 1 - \frac{\delta'2^{-d}}{2\sqrt{d}d(M+1)^2}$. Hence, w.p. at least $1 - \delta'$, the distance is at least $\sqrt{\frac{\delta'2^{-d}}{2\sqrt{d}d(M+1)^2}} \geq \frac{\delta'2^{-d}}{2\sqrt{d}d(M+1)^2}$.

Hence, we get for each pair of critical hyperplanes a distance of at least $\delta$ w.p. $1 - p_5 = 1 - \delta q^2\sqrt{d}d(M+1)^22^{d+1}$.

6. The angle between the $j$th neuron and the axis $\{t\boldsymbol{e}^{(i)} : t \in \mathbb{R}\}$ equals to $\left|\langle\mathbf{w}'_j, \boldsymbol{e}^{(i)}\rangle\right| = \left|\mathrm{w}'_{j,i}\right|$. The probability for this to be at least $\delta$ is

$$p\left(\left|\mathrm{w}'_{j,i}\right| > \delta\right) \geq p(|\mathrm{r}_{j,i}| > \delta) = 1 - \frac{\delta}{2^{-d}}.$$

Using the union bound, we have that the probability for each neuron and each axis to have an angle of at least $\delta$ is $1 - p_6$, where $p_6 = dq\delta 2^d$.

$\square$

**Lemma 7.** *Let $\mathcal{M}$ be a three-layers neural network as in equation 3. Let $q$ be the number of neurons in the network, and let $M$ be an upper bound on the absolute value of the weights. For each weight in the network, add a uniform element in $[-2^{-d}, 2^{-d}]$, and denote the the noisy network by $\mathcal{M}'$. Then:*

1. *For each $i \in [d]$, all critical points of the piecewise linear function $t \mapsto \mathcal{M}(t\boldsymbol{e}^{(i)})$ are in $\left(-\frac{1}{\delta}, \frac{1}{\delta}\right) \setminus (-\delta, \delta)$ with probability $1 - p_1 = 1 - dq^2\delta(dM^2 + 2)2^{d+1}$.*

2. *For each $i \in [d]$, each piece in the piecewise linear function $t \mapsto \mathcal{M}(t\boldsymbol{e}^{(i)})$ is of length at least $\delta$ with probability $1 - p_2 = 1 - d2^d q^4\delta(dM^2 + 2)$.*

3. *For each $i \in [d]$, all the points in the grid $\frac{2^{-\lceil\log_2(2/\delta^2)\rceil}}{\delta}\mathbb{Z}$ of the piecewise linear function $t \mapsto \mathcal{M}(t\boldsymbol{e}^{(i)})$ are $\delta^2$-general with probability $1 - p_3 = 1 - 3dq^2\delta$.*

4. *Any critical point in the axes $\{t\boldsymbol{e}^{(i)} : t \in \mathbb{R}\}$ is $\delta$-non-degenerate with probability $1 - p_4 = 1 - d^{3/2}q^4\delta(dM^2 + 2)2^{d-1}$.*

5. *The distance between each pair of critical hyperplanes is at least $\delta$ with probability $1 - p_5 = 1 - \delta q^4\sqrt{d}d(dM^2 + 2)^22^{d+1}$.*

6. *The angle between any critical hyperplane and any axis is at least $\delta$ with probability $1 - p_6 = 1 - dq^2\delta 2^d$.*

7. *In the case of depth-three networks, we assume that the above conditions also apply to the sub-network defined by the top two layers with probability $1 - p_7$ where $p_7$ is the sum of the probabilities of lemma 6.*

*Proof.* Lemma 5 tells us that each non-zero critical hyperplane $\mathbb{P}$ corresponding to a second layer neuron also corresponds to a single first layer state almost surely. Therefore, given $i \in [d]$, we can consider the $q'$ critical points that intersect with the $i$th axis as $q'$ first-layer neurons, where each second neuron is multiplied by an affine transformation that is the current state of the first neurons. As each first layer neuron intersects with the axis at most once, and each second layer neuron intersects with the axis at most $q_1$ times, where $q_1$ is the number of first layer neurons, we can bound $q'$ by $q' \leq q^2$.

Furthermore, given a critical hyperplane $\mathbb{P}$ corresponding to a second layer neuron $j$, denote by $(\mathbf{W}'_{\mathbb{P}}, \mathbf{b}'_{\mathbb{P}})$ the state of that first layer (which is the same as $(\mathbf{W}', \mathbf{b}')$ as defined in the proof of Lemma 6, except to some zero rows due to ReLU). That is, $\mathbb{P} = \{\boldsymbol{x} : \langle \boldsymbol{v}_j, \mathbf{W}'_{\mathbb{P}}\boldsymbol{x}\rangle + \langle \boldsymbol{v}_j, \mathbf{b}'_{\mathbb{P}}\rangle + c_j = 0\}$ which can be viewed locally as a pseudo-neuron with parameters $((\mathbf{W}'_{\mathbb{P}})^T\boldsymbol{v}_j, +\langle \boldsymbol{v}_j, \mathbf{b}'_{\mathbb{P}}\rangle + c_j)$ that are each bounded in magnitude by $M' \leq dM^2 + 1$.

1. Applying the above to lemma 6, we get:
$$p_1 \leq dq'\delta(M' + 1)2^{d+1} \leq dq^2\delta(dM^2 + 2)2^{d+1}.$$

2. Applying the above to lemma 6, we get:
$$p_2 \leq d2^d q'^2\delta(M' + 1) \leq d2^d q^4\delta(dM^2 + 2).$$

3. Applying the above to lemma 6, we get:
$$p_3 \leq 3dq'\delta \leq 3dq^2\delta.$$

4. Applying the above to lemma 6, we get:
$$p_4 \leq d^{3/2}q'^2\delta(M' + 1)2^{d-1} \leq d^{3/2}q^4\delta(dM^2 + 2)2^{d-1}.$$

5. Note that the maximal possible number of critical hyperplanes is at most $h = q^2$, as interactions between each first-layer neuron and a second-layer neuron may cause a single hyperplane. Therefore, we get:
$$p_5 = \delta h^2\sqrt{d}d(M' + 1)^2 2^{d+1} \leq \delta q^4\sqrt{d}d(dM^2 + 2)^2 2^{d+1}.$$

6. Applying the above to lemma 6, we get:
$$p_6 \leq dq'\delta 2^d \leq dq^2\delta 2^d.$$

7. Let $p_1, p_2, p_3, p_4, p_5, p_6$ as defined on lemma 6. As the number of neurons in the second layer is at most $q$, using the union bound we get: $p_7 = \sum_{i=1}^{6} p_i$.

$\square$

In the rest of the section, we prove lemmas stated in section 3.3.

*Proof.* (of lemma 1) The proof follow from lemma 5 and the fact that the number of neurons - hence, the number of critical hyperplanes - is finite. $\square$

*Proof.* (of lemma 2) From lemma 1, we have that the perturbed network $\mathcal{M}'$ is regular almost surely. This implies that it is $\delta$-regular for some $\delta > 0$, As the number of neurons is finite.

Fix a $\delta > 0$. For two-layer networks, lemma 6 bounds the probability to dispose one of the restrictions of $\delta$-regular network. Let $p_1, \ldots, p_6$ as in lemma 6, then, using the union bound, we get that the network is $\delta$-regular with probability of at least $1 - p$ where

$$p = p_1 + p_2 + p_3 + p_4 + p_5 + p_6$$
$$= dq\delta(M + 1)2^{d+1} + d2^d q^2\delta(M + 1) + 3dq\delta$$
$$+ d^{3/2}q^2\delta(M + 1)2^{d-1} + \delta q^2\sqrt{d}d(M + 1)^2 2^{d+1} + dq\delta 2^d$$
$$< 10(M + 1)^2 q^2 d^{3/2}\delta 2^d.$$

Therefore, if we choose $\delta = (10(M + 1)^2 q^2 d^{3/2} 2^{2d})^{-1}$ we will get the requested bound.

For three-layer networks, let $p_1, \ldots, p_7$ as in lemma 7, then, using the union bound, we get that the network is $\delta$-regular with probability of at least $1 - p'$ where

$$
\begin{aligned}
p' &= p_1 + p_2 + p_3 + p_4 + p_5 + p_6 + p_7 \\
&\leq dq^2\delta(dM^2 + 2)2^{d+1} + d2^d q^4 \delta(dM^2 + 2) + 3dq^2\delta + d^{3/2}q^4\delta(dM^2 + 2)2^{d-1} \\
&\quad + \delta q^4 \sqrt{d}d(dM^2 + 2)^2 2^{d+1} + dq^2\delta 2^d + 10(M + 1)^2 q^2 d^{3/2}\delta 2^d \\
&< 20(dM^2 + 2)^2 q^4 d^{3/2}\delta 2^d.
\end{aligned}
$$

Therefore, if we set $\delta = (20(dM^2 + 2)^2 q^4 d^{3/2} 2^{2d})^{-1}$ we will get the requested bound. $\qquad\square$

*Proof.* (of lemma 3) Let $\mathbf{W} \in \mathbb{R}^{d_1 \times d}$ a random matrix, where each element is drawn independent of the other, and define by $\mathbf{w}_j$ its $j$'th row, $j \in [d]$. Also, let $r = \min\{d, d_1\}$. Note that $\mathbf{W}$ has a full rank with probability 1, where by full rank we mean that $\mathrm{rank}(\mathbf{W}) = \min\{d, d_1\} = r$. Indeed, consider drawing at random the $j$th row, for $j \leq r$ after fixing the first $j - 1$ rows. In order of that row to be dependent in $\mathbf{w}_1, \ldots, \mathbf{w}_{j-1}$, then $\mathbf{w}_j$ must fall in a subspace whose dimension is at most $j - 1 < r$, which has a zero Lebesgue measure in an $r$-dimensional space.

Therefore, if $d_1 \leq d$ then $r = d_1$ and $W$ has a rank $d_1$ with probabilty 1. The Rank–nullity theorem then implies that the image of $\mathbf{W}$ is a $d_1$-dimensional space, and thus $\mathbf{W}$ is surjective. $\qquad\square$

*Proof.* (of lemma 4) The proof follows from Theorem 5. If we set $3.5d_1 \leq d_2$ we get a probability of:

$$
1 - \frac{\left(\frac{ed_2}{d_1}\right)^{d_1+1}}{2^{d_2}} \leq 1 - \frac{(3.5e)^{d_1+1}}{2^{3.5d_1}} = 1 - 3.5e\left(\frac{3.5e}{2^{3.5}}\right)^{d_1} \xrightarrow{d_1 \to \infty} 1.
$$

$\qquad\square$

# B    PROOF OF THE MAIN THEOREMS

*Proof.* (of theorem 1) The correctness of the theorem follows from the correctness of theorem 4 below. $\qquad\square$

*Proof.* (of theorem 4) We will assume without loss of generality that the $u_i$'s are in $\{\pm 1\}$, as any neuron $\boldsymbol{x} \mapsto u\phi(\langle \boldsymbol{w}, \boldsymbol{x}\rangle + b)$ calculates the same function as $\boldsymbol{x} \mapsto \frac{u}{|u|}\phi(\langle |u|\boldsymbol{w}, \boldsymbol{x}\rangle + |u|b)$, as ReLU is a positive homogeneous function.

Let $\mathbb{S} = \{\boldsymbol{x}_1, \ldots, \boldsymbol{x}_m\}$ be the list of points found using FIND_CP in algorithm 6. Our general assumption is that all the critical points on the line $\mathbb{R}\boldsymbol{e}^{(1)}$ are on the range $\left(-\frac{1}{\delta}, \frac{1}{\delta}\right)\boldsymbol{e}^{(1)}$. Hence, from the correctness of lemma 13, we are guarantees that all the critical points on the line $\mathbb{R}\boldsymbol{e}^{(1)}$ are in $\mathbb{S}$. We claim that for each $\boldsymbol{x} \in \mathbb{S}$ there is exactly one critical hyperplane $\mathbb{P}$ with $\boldsymbol{x} \in \mathbb{P}$, and $|\mathbb{S} \cap \mathbb{P}| = 1$. Assume by contradiction that one of the above is false. If $\boldsymbol{x} \notin \mathbb{P}$ for all the critical hyperplanes, then $\boldsymbol{x}$ is not a critical point, which contradicts lemma 13. If $|\mathbb{S} \cap \mathbb{P}| = 0$ this means that $\mathbb{P}$ does not intersect with $\boldsymbol{e}^{(1)}$, i.e., parallel to this axis, which contradict our general position assumption. Finally, if $|\mathbb{S} \cap \mathbb{P}| > 1$ this means that $\mathbb{P}$ intersects with $\boldsymbol{e}^{(1)}$, which means $\mathbb{P}$ is not affine. Therefore, each neuron is represented by a unique critical point $\boldsymbol{x} \in \mathbb{S}$.

Let $\boldsymbol{x} \in \mathbb{S}$ be a critical point of the $j$'th neuron, and $(\boldsymbol{w}', b') = $ FIND_HP$(\mathcal{M}, \delta, \boldsymbol{x})$. From lemma 14 we get that either $(\boldsymbol{w}_j, b_j) = (\boldsymbol{w}', b')$ or $(\boldsymbol{w}', b') = (-\boldsymbol{w}'_j, -b'_j)$. To recover $u_j$, note that if $u_j = 1$ then $\mathcal{M}(x)$ is strictly convex in $\mathbb{B}(\boldsymbol{x}, \delta)$ as the sum of the affine function $\mathcal{M}'(\boldsymbol{x})$ and the convex function $u_j\phi(\langle \boldsymbol{w}_j, \boldsymbol{x}\rangle + b_j)$. Similarly, if $u_j = -1$ then $\mathcal{M}(x)$ is strictly concave in $\mathbb{B}(\boldsymbol{x}, \delta)$. Thus, using algorithm 4, we will be able to determine $u_j$ correctly.

Let $\mathbb{C} \subset [d_1]$ be the set of neurons assigned to an incorrect sign. Then, for all $\boldsymbol{x} \in \mathbb{R}^d$:

$$\mathcal{M}(x) - \sum_{j=1}^{d_1'} u_j' \phi\left(\langle \boldsymbol{w}_j', \boldsymbol{x} \rangle + b_j'\right) = \sum_{j \in \mathbb{C}} u_j \phi\left(\langle \boldsymbol{w}_j, \boldsymbol{x} \rangle + b_j\right) - u_j \phi\left(\langle \boldsymbol{w}_j, \boldsymbol{x} \rangle - b_j\right)$$
$$= \sum_{j \in \mathbb{C}} u_j\left(\langle \boldsymbol{w}_j, \boldsymbol{x} \rangle + b_j\right)$$

which is an affine transformation and can be recovered successfully at the last stage of the algorithm.

As for the time and query complexity, step 1 takes $O\left(d_1 \log(1/\delta)Q\right)$ (see section 3.2). Since each neuron correspond to a single critical point, we have that $m = d_1$. Thus the loop in step 2 makes $d_1$ iterations. The cost of each iteration is $O(dQ)$. Hence, the total cost of the loop is $O(d_1 dQ)$. Finally, to perform step 6 we need to make $d$ queries to $\mathcal{M}$ which cost $O(dQ)$, and also $d$ evaluations of $\sum_{i=1}^m u_i' \phi\left(\Lambda_{\boldsymbol{w}_i', b_i'}(\boldsymbol{x})\right)$ which cost $d_1 d$ each. The total runtime is therefore $O\left(d_1 \log(1/\delta)Q + d_1 dQ + dQ + d^2 d_1\right) = O\left(d_1 \log(1/\delta)Q + d_1 dQ + d^2 d_1\right)$. $\qquad\square$

*Proof.* (of theorem 2) Denote the output of the $j$'th neuron before the activation by $\mathcal{M}_j(\boldsymbol{x}) = \boldsymbol{w}_j \boldsymbol{x} + b_j$.

Let $\boldsymbol{x}_1, \boldsymbol{x}_2 \in \mathbb{R}_+^d$ be two points such that exactly one neuron $j \in [d_1]$ changed its state (i.e. changed from active to inactive, or vice versa) in the segment $[\boldsymbol{x}_1, \boldsymbol{x}_2] := \{\lambda \boldsymbol{x}_1 + (1 - \lambda)\boldsymbol{x}_2 : \lambda \in [0,1]\}$.

Moreover, assume that no neuron changes its state in neighborhoods of $\boldsymbol{x}_1$ and $\boldsymbol{x}_2$, so that the change in the state happens in the interior of $[x_1, x_2]$. We note that finding such a pair of points $\boldsymbol{x}_1, \boldsymbol{x}_2$ can be done by considering a ray $\ell(\rho) := \rho e^{(i)}$, and seeking a critical point $\tilde{\rho}$ of the (one dimensional) function $N \circ \ell$. Under our general position assumptions, for some $j \in [d]$, there is such a $\rho$ in $\mathbb{R}$, and it can be found efficiently. Given such a $\rho$, and again under our general position assumptions, we can take $\boldsymbol{x}_1 = \ell(\rho - \epsilon)$ and $\boldsymbol{x}_2 = \ell(\rho + \epsilon)$, for small enough $\epsilon$.

We will explain next how given such two points, we can reconstruct the $j$'th neuron, up to an affine function. First, the reconstruction of $u_j$ is simple. Indeed, in the segment $[x_1, x_2]$, $\mathcal{M}'(\boldsymbol{x}) := \mathcal{M}(\boldsymbol{x}) - u_j \phi(\boldsymbol{w}_j \boldsymbol{x} + b_j)$ is affine, as no neuron, except the $j$'th neuron, changes its mode. Hence, $\mathcal{M}(\boldsymbol{x}) = u_j \phi(\boldsymbol{w}_j \boldsymbol{x} + b_j) + \mathcal{M}'(\boldsymbol{x})$ is a sum of an affine function and the $j$'th neuron. In particular, it is convex iff the $j$'th neuron is convex iff $u_j = 1$. Hence, to reconstruct $u_j$ we only need to check if the restriction of $N$ to $[x_1, x_2]$ is convex or concave.

We next explain how to reconstruct an affine map $\Lambda$ such that $\phi(\Lambda(\boldsymbol{x})) - \phi(\mathcal{M}_j(\boldsymbol{x}))$ is affine. Let $\Lambda_1, \Lambda_2 : \mathbb{R}^d \to \mathbb{R}$ be the affine maps computed by the networks in the neighborhoods of $\boldsymbol{x}_1$ and $\boldsymbol{x}_2$ respectively. Note that it is straight forward to reconstruct $\Lambda_i$ from the set $\mathcal{M}(x_i), \mathcal{M}(x_i + \epsilon e_1), \ldots, \mathcal{M}(x_i + \epsilon e_d)$, for small enough $\epsilon$. We have that $\Lambda := \Lambda_1 - \Lambda_2$ is either $N_j$ or $-N_j$. Hence, we have that $\phi(\Lambda(\boldsymbol{x}))$ is either $\phi(\mathcal{M}_j(\boldsymbol{x}))$ or $\phi(-\mathcal{M}_j(\boldsymbol{x})) = \phi(\mathcal{M}_j(\boldsymbol{x})) - \mathcal{M}_j(\boldsymbol{x})$.

After removing all the neurons, we are left with an affine map that can be reconstructed easily using $O(d)$ queries as explained above, and the full reconstruction of the network is complete.

$\qquad\square$

*Proof.* (of theorem 3) Recall that our goal is to recover a $\delta$-regular network of the form

$$\mathcal{M}(\boldsymbol{x}) = \langle \boldsymbol{u}, \phi(\boldsymbol{V}\phi(\boldsymbol{W}\boldsymbol{x} + \boldsymbol{b}) + \boldsymbol{c})\rangle.$$

We denote by $\boldsymbol{w}_j$ the $j$th row of $\boldsymbol{W}$ and assume without loss of generality that it is of unit norm, as any neuron of the form $\boldsymbol{x} \mapsto \phi(\langle \boldsymbol{w}, \boldsymbol{x} \rangle + b)$ can be replaced by $\boldsymbol{x} \mapsto \|\boldsymbol{w}\|\phi\left(\left\langle \frac{\boldsymbol{w}}{\|\boldsymbol{w}\|}, \boldsymbol{x} \right\rangle + \frac{b}{\|\boldsymbol{w}\|}\right)$. Likewise, and similar to our algorithm for reconstruction of depth-two networks, we will assume that $\boldsymbol{u} \in \{\pm 1\}^{d_2}$.

The first step of the algorithm would be to find a list

$$L = \left\{(\boldsymbol{x}_1, \hat{\mathbb{P}}_1), \ldots, (\boldsymbol{x}_m, \hat{\mathbb{P}}_m)\right\}$$

of pairs such that

- For each $k$, $\mathbb{P}_k$ is a critical hyperplane of $\mathcal{M}$ and $\boldsymbol{x}_k$ is a $\delta$-non-degenerate critical point whose critical hyperplane is $\mathbb{P}_k$

- The list contains all the critical hyperplanes of first-layer neurons

For that we will use repeatedly FIND_CP$(\delta, t \mapsto \mathcal{M}(t\boldsymbol{e}^{(1)}), \cdot)$ to find all the critical points on the axis $\{t\boldsymbol{e}^{(1)} : t \in \mathbb{R}\}$ (see section 3.2), similar to our algorithm for reconstructing depth two networks. Denote those set of points by $\mathbb{S} = \{\boldsymbol{x}_1, \dots, \boldsymbol{x}_m\}$. Lemma 13 along with the general position assumption, guarantee that for each critical hyperplane $\mathbb{P}$ that corresponds to a first-layer neuron, $|\mathbb{P} \cap \mathbb{S}| = 1$, and that all the points in $\mathbb{S}$ are $\delta$-non-degenerate. Then, using algorithm 3 we will find the critical hyperplane $\mathbb{P}_k$ for each point $\boldsymbol{x}_k \in \mathbb{S}$.

For the runtime, note that $m = O(d_1 d_2)$ (e.g. Telgarsky (2016)), the critical points can be found in time $O\left(d_1 d_2 \log(1/\delta) Q\right)$ as explained in section 3.2.2, and each hyperplane $\hat{\mathbb{P}}_i$ can be efficiently found via $O(d)$ queries near $\boldsymbol{x}_i$ as explained in section 3.2.3. The total running time of this step is therefore $O\left(d_1 d_2 \log(1/\delta) Q + d_1 d_2 d Q\right)$.

The second step is to take the list $L = \left\{(\boldsymbol{x}_1, \hat{\mathbb{P}}_1), \dots, (\boldsymbol{x}_m, \hat{\mathbb{P}}_m)\right\}$ and remove all the points that don't correspond to first-layer neurons. After that, the list will contain precisely the critical hyperplanes of the neurons in the first layer. In order to do so, it is enough to efficiently decide, given the list $L$, whether a given hyperplane $\mathbb{P}$ is a critical hyperplane of a neuron in the first layer. The idea behind this verification is simple: If $\mathbb{P}$ corresponds to a neuron in the first layer then any point in $\mathbb{P}$ is a critical point of $\mathcal{M}$ (see lemma 8). Indeed, suppose that $\mathbb{P}$ is critical at $\mathbb{P}$ for a first layer neuron $h(\boldsymbol{x}) = \phi(\boldsymbol{w}\boldsymbol{x} + b)$. We have that $\mathbb{P}$ is the null space of the affine input to $h$ in the proximity of $\mathbb{P}$. But the input to $h$ is the same affine function in the proximity of every point $\boldsymbol{x} \in \mathbb{R}^d$. Thus, for every $\boldsymbol{x} \in \mathbb{R}^d$ is a critical point for $h$ with $\mathbb{P}$ as its critical hyperplane. On the other hand, if $\mathbb{P}$ corresponds to a neuron in the second layer, then not all its points are critical for $\mathcal{M}$: Indeed, suppose that we start from $\boldsymbol{x} \in \mathbb{P}$, which is critical for $\mathcal{M}$ and start to move inside $\mathbb{P}$ until one of the neurons in the first layer changes its state. Then we will reach a point in $x' \in \mathbb{P}$, which is not critical for $\mathcal{M}$, as, by our general position assumption, $\mathbb{P}$ corresponds to a single first layer state. Thus, in order to verify if $\mathbb{P}$ corresponds to a first layer neuron, we will go over all the hyperplanes $\hat{\mathbb{P}}_k \in L$, and for each of them, will find a point $\boldsymbol{x}' \in \mathbb{P}$ that is the opposite side of $\hat{\mathbb{P}}_k$ (relative to $\boldsymbol{x}$) and will check if it is critical. If it is not critical for one of the hyperplanes, we know that $\mathbb{P}$ does not correspond to a first layer neuron. If all the points that we have examined are critical, even for $\hat{\mathbb{P}}_k$ corresponding to a first layer neuron, then $\boldsymbol{x}'$ is critical, which means that $\mathbb{P}$ must correspond to a first layer neuron.

Algorithm 7 implements this idea. There is one caveat that we need to handle: The examined point has to be generic enough in order to test whether it is critical or not using algorithm 5. To make sure that the point is general enough, we slightly perturb it. The correctness of the algorithm follows from lemmas 10 and 11. Indeed, if $\mathbb{P}$ corresponds to a first layer neuron, then lemma 10 implies that each test in the for loop will fail w.p. at least $1 - \frac{2^{-d}}{m^2}$. Thus, w.p. at least $1 - \frac{2^{-d}}{m}$ all the tests will fail, and the algorithm will reach step 8 and will correctly output that "$\mathbb{P}$ is a first-layer critical hyperplane." In the case that $\mathbb{P}$ corresponds to a second layer neuron, lemma 11 implies that once we will reach an iteration in which $\hat{\mathbb{P}}_k$ corresponds to a first layer neuron, the test in step 6 will succeed w.p at least $1 - \frac{2^{-d}}{m^2}$, in which case the algorithm will correctly output "$\mathbb{P}$ is not a first-layer critical hyperplane." All in all, it follows that the algorithm will output the correct output w.p. at least $1 - \frac{2^{-d}}{m}$ for every hyperplane $\mathbb{P}$. Thus, w.p. at least $1 - 2^{-d}$ it will output the correct answer for all hyperplanes.

As for runtime, note that each step in the for-loop takes $O(dQ)$. As the list size is $O(d_1 d_2)$, the total running time over all hyperplanes is $O(d_1^2 d_2^2 d Q)$.

Since the rows in $\boldsymbol{W}$ are assumed to have a unit norm, the list of the critical hyperplanes of the first-layer neurons, obtained in the previous step, determines the weights up to sign. Namely, we can reconstruct a list

$$L = \left\{(\hat{\boldsymbol{w}}_1, \hat{b}_1), \dots, (\hat{\boldsymbol{w}}_{d_1} \hat{b}_{d_1})\right\}$$

that define precisely the neurons on the first layer, up so sign. For the third step, it, therefore, remains to recover the correct signs (note that this process is only required for inner layers and avoidable for the top layer, as explained above).

In order to recover the correct sign of $(\hat{\boldsymbol{w}}_1, \hat{b}_1)$, we can simply do the following test: Choose a point $\boldsymbol{x}$ such that $\hat{\boldsymbol{w}}_1 \boldsymbol{x} + \hat{b}_1 = 0$, and query the network in the points $\boldsymbol{x} + \epsilon \boldsymbol{z}, \boldsymbol{x} - \epsilon \boldsymbol{z}$, for small $\epsilon$, where $\boldsymbol{z} \in \mathbb{R}^d$ is a unit vector that has the property that is orthogonal to $\hat{\boldsymbol{w}}_2, \ldots, \hat{\boldsymbol{w}}_{d_1}$, but $\hat{\boldsymbol{w}}_1 \boldsymbol{z} > 0$. If we assume that $W$ is right invertible, then such a $\boldsymbol{z}$ exists, as $\boldsymbol{w}_1, \ldots, \boldsymbol{w}_{d_1}$ are linearly independent.

Now, when moving from $\boldsymbol{x}$ to either $\boldsymbol{x} + \epsilon \boldsymbol{z}$ or $\boldsymbol{x} - \epsilon \boldsymbol{z}$, the value of all the neurons in the first layer, possibly except the one that corresponds to $(\hat{\boldsymbol{w}}_1, \hat{b}_1)$, does not change. As for the neuron that corresponds to $(\hat{\boldsymbol{w}}_1, \hat{b}_1)$, if its real weights are indeed $(\hat{\boldsymbol{w}}_1, \hat{b}_1)$, then its value changes when we move from $\boldsymbol{x}$ to $\boldsymbol{x} + \epsilon \boldsymbol{z}$ but not when we move from $\boldsymbol{x}$ to $\boldsymbol{x} - \epsilon \boldsymbol{z}$. On the other hand, if its real weights are $(-\hat{\boldsymbol{w}}_1, -\hat{b}_1)$, then the value changes when we move from $\boldsymbol{x}$ to $\boldsymbol{x} - \epsilon \boldsymbol{z}$ but not when we move from $\boldsymbol{x}$ to $\boldsymbol{x} + \epsilon \boldsymbol{z}$. Hence, in order to have the correct sign we can do the following: If $\mathcal{M}(\boldsymbol{x}) \neq \mathcal{M}(\boldsymbol{x} + \epsilon \boldsymbol{z})$ then keep $(\hat{\boldsymbol{w}}_1, \hat{b}_1)$. Else, replace it with $(-\hat{\boldsymbol{w}}_1, -\hat{b}_1)$. This test works because of the above discussion, together with the assumption that the second layer has non-zero partial derivatives; therefore, we can guarantee that either $\boldsymbol{x} + \epsilon \boldsymbol{z}$ or $\boldsymbol{x} - \epsilon \boldsymbol{z}$ will show a change in the values of $\mathcal{M}$. More on the non-zero partial derivatives assumption, see section E.

The runtime of this step is $O(d_1^3 d + d_1 Q)$. Indeed, to find $\boldsymbol{z}$, we need to do Gram-Schmidt, which takes $O(d_1^2 d)$. After that, all that is needed is two queries. We need to do this for each first layer neuron, so the total runtime is $O(d_1^3 d + d_1 Q)$.

For the fourth step, we shall recover the values of the top layer up to an affine transformation. After having the weights of the first layer at hand, and since $W$ is assumed to be right invertible, we can directly access the sub-network defined by the top two layers. Namely, given $\boldsymbol{x} \in \mathbb{R}_+^{d_1}$, we can find $\boldsymbol{z} \in \mathbb{R}^d$ such that

$$\boldsymbol{x} = (\phi(\boldsymbol{w}_1 \boldsymbol{z} + b_1), \ldots, \phi(\boldsymbol{w}_{d_1} \boldsymbol{z} + b_{d_1}))$$

e.g., by taking $\boldsymbol{z} = \boldsymbol{W}^{-1}(\boldsymbol{x} - \boldsymbol{b})$ where $\boldsymbol{W}^{-1}$ is a right inverse of $\boldsymbol{W}$. Now, $\mathcal{M}(\boldsymbol{z})$ is precisely the value of the top layer on the input $\boldsymbol{x}$. Hence, the problem of reconstructing the top two layers boils down to the problem of reconstructing a depth two network in the $\mathbb{R}_+^d$-restricted case, which its correctness is given in theorem 2.

The cost of a query to the second layer is $Q$ plus the cost of computing $\boldsymbol{z}$, which is $O(dd_1)$. There is also an asymptotically negligible cost of $O(d_1^2 d)$ for computing $\boldsymbol{W}^{-1}$. The runtime of this step is therefore $O\big((\log(1/\delta) + d_1) d_1 d_2 (Q + dd_1) + d_1^2 d_2^2\big)$.

$\square$

## C  PROOFS OF LEMMAS

**Lemma 8.** *Let $\mathbb{P}$ be a critical hyperplane corresponding to a first layer neuron. Then, any point in $\mathbb{P}$ is critical for $\mathcal{M}$.*

*Proof.* W.l.o.g. $\mathbb{P}$ corresponds to the neuron $\phi(\boldsymbol{w}_1 \boldsymbol{x} + b_1)$. Let $\boldsymbol{x}_0 \in \mathbb{P}$ and let $\boldsymbol{e}$ be a unit vector that is orthogonal to $\boldsymbol{w}_2, \ldots, \boldsymbol{w}_{d_1}$ and such that $\langle \boldsymbol{w}_1, \boldsymbol{e} \rangle > 0$. Such $\boldsymbol{e}$ exists as we assume that $\boldsymbol{w}_1, \ldots, \boldsymbol{w}_{d_1}$ are independent.

Consider the function $f(t) = \mathcal{M}(\boldsymbol{x}_0 + t\boldsymbol{e})$. We claim that it is not linear in any neighborhood of 0, which implies that $\boldsymbol{x}_0$ is critical. Indeed, for all $i > 1$, $t \mapsto \phi(\boldsymbol{w}_i(\boldsymbol{x}_0 + t\boldsymbol{e}) + b_i)$ is constant, as $\boldsymbol{e}$ is orthogonal to $\boldsymbol{w}_i$. As for $i = 1$, $t \mapsto \phi(\boldsymbol{w}_1(\boldsymbol{x}_0 + t\boldsymbol{e}) + b_1)$ is the zero function for $t \leq 0$, as in this case $\boldsymbol{w}_1(\boldsymbol{x}_0 + t\boldsymbol{e}) + b_1 < \boldsymbol{w}_1 \boldsymbol{x}_0 + b_1 = 0$. Hence, the left derivative of $f$ at 0 is 0. On the other hand, for $t > 0$, $\phi(\boldsymbol{w}_1(\boldsymbol{x}_0 + t\boldsymbol{e}) + b_1) = \boldsymbol{w}_1(\boldsymbol{x}_0 + t\boldsymbol{e}) + b_1 = t\boldsymbol{w}_1 \boldsymbol{e}$. Hence, the right derivative of $g(t) = \phi(\boldsymbol{W}(\boldsymbol{x}_0 + t\boldsymbol{e}) + \boldsymbol{b})$ is $\langle \boldsymbol{w}_1, \boldsymbol{e} \rangle \boldsymbol{e}^{(1)}$. Now, it is assumed that the derivative of $F(\boldsymbol{z}) = \boldsymbol{u}\phi(V\boldsymbol{z} + \boldsymbol{c})$ in the direction of $\boldsymbol{e}^{(1)}$ is not zero. Hence, the right derivative of $f(t) = F(g(t))$ is not zero. All in all we have shown that the right derivative of $f$ at 0 is different from the left derivative, which implies that $f$ is not linear in any neighborhood of 0. $\square$

**Lemma 9.** *Let $\mathbb{P}_1, \mathbb{P}_2$ be hyperplanes such that $D(\mathbb{P}_1, \mathbb{P}_2) \geq \delta$. Let $\boldsymbol{x} \in \mathbb{P}_1$ and let $\mathbf{x}$ be a standard Gaussian in $\mathbb{P}_1$ with mean $\boldsymbol{x}$. Then $p\left(d(\mathbf{x}, \mathbb{P}_2) \leq a\right) \leq \frac{\sqrt{2}a}{\delta\sqrt{\pi}}$.*

*Proof.* W.l.o.g. we can assume that $\mathbb{P}_1$ and $\mathbb{P}_2$ contain the origin. Let $\boldsymbol{n}_2$ be the normal of $\mathbb{P}_2$. We have that $d(\mathbf{x}, \mathbb{P}_2) = |\langle \mathbf{x}, \boldsymbol{n}_2 \rangle|$. Now

$$
\begin{aligned}
\langle \mathbf{x}, \boldsymbol{n}_2 \rangle &= \langle \mathbf{x}, \boldsymbol{n}_2 - \mathrm{proj}_{\mathbb{P}_1} \boldsymbol{n}_2 \rangle + \langle \mathbf{x}, \mathrm{proj}_{\mathbb{P}_1} \boldsymbol{n}_2 \rangle \\
&\overset{\mathbf{x} \in \mathbb{P}_1}{=} \langle \mathbf{x}, \mathrm{proj}_{\mathbb{P}_1} \boldsymbol{n}_2 \rangle \\
&= \langle \mathbf{x} - \boldsymbol{x}, \mathrm{proj}_{\mathbb{P}_1} \boldsymbol{n}_2 \rangle + \langle \boldsymbol{x}, \mathrm{proj}_{\mathbb{P}_1} \boldsymbol{n}_2 \rangle
\end{aligned}
$$

Hence $\langle \mathbf{x}, \boldsymbol{n}_2 \rangle$ is a Gaussian with mean $\mu := \langle \boldsymbol{x}, \mathrm{proj}_{\mathbb{P}_1} \boldsymbol{n}_2 \rangle$ and variance $\phi^2 \geq \delta^2$. Hence,

$$
p(\mathbf{x} \in [-a, a]) = \frac{1}{\phi\sqrt{2\pi}} \int_{-a}^{a} e^{-\frac{1}{2}\left(\frac{t-\mu}{2}\right)^2} dt \leq \frac{2a}{\delta\sqrt{2\pi}} = \frac{\sqrt{2}a}{\delta\sqrt{\pi}}.
$$

$\square$

**Lemma 10.** *Let $\mathbb{P}$ be a hyperplane that corresponds to a first layer neuron. Let $\boldsymbol{x} \in \mathbb{P}$ and let $\mathbf{x}$ be a standard Gaussian in $\mathbb{P}$ with mean $\boldsymbol{x}$. Then $\mathbf{x}$ is $\delta'$-non-degenerate critical point of $\mathbb{P}$ w.p. at least $1 - 2^{2(d_1+d_2)} \frac{\delta'\sqrt{2}}{\delta\sqrt{\pi}}$.*

*Proof.* By lemma 8 $\mathbf{x}$ is critical w.p. 1. It is therefore enough to show that w.p. at least $1 - 2^{2(d_1+d_2)} \frac{\delta'\sqrt{2}}{\delta\sqrt{\pi}}$, the distance of $\mathbf{x}$ from every critical hyperplane other than $\mathbb{P}$ is at least $\delta'$. Indeed, by lemma 9 and the fact that there are at most $(d_1 + d_2)2^{d_1+d_2} \leq 2^{2(d_1+d_2)}$ critical hyperplanes, the probability that the distance from $\mathbf{x}$ to one of the critical hyperplane is less than $\delta'$ is at most $2^{2(d_1+d_2)} \frac{\delta'\sqrt{2}}{\delta\sqrt{\pi}}$. $\square$

**Lemma 11.** *Let $\mathbb{P}$ be a hyperplane that corresponds to a second layer neuron. Let $\boldsymbol{x}_1 \in \mathbb{P}$ be a critical point with $\mathbb{P}$ as its critical hyperplane. Let $\mathbb{P}_1$ be a hyperplane that corresponds to a first layer neuron. Let $x_2 \in \mathbb{P}$ be another point and assume that $\boldsymbol{x}_1$ and $\boldsymbol{x}_2$ are of opposite sides of $\mathbb{P}_1$. Let $\mathbf{x}$ be a standard Gaussian in $\mathbb{P}$ with mean $\boldsymbol{x}_2$. Then $\mathbf{x}$ is $\delta'$-general w.p. at least $1 - 2^{2(d_1+d_2)} \frac{\delta'\sqrt{2}}{\delta\sqrt{\pi}} - e^{-\frac{(d(\boldsymbol{x}_2, \mathbb{P}_1) - \delta')^2}{2}}$.*

*Proof.* As in the proof of lemma 10 the probability that the distance from $\mathbf{x}$ to one of the critical hyperplanes other than $\mathbb{P}$ is less than $\delta'$ is at most $2^{2(d_1+d_2)} \frac{\delta'\sqrt{2}}{\delta\sqrt{\pi}}$. It is therefore remains to show that the probability that $\mathbf{x}$ is $\delta'$-close to on of $\mathbb{P}$'s critical points is at most $e^{-\frac{(d(\boldsymbol{x}_2, \mathbb{P}_1) - \delta')^2}{2}}$.

Denote by $\boldsymbol{n}_1$ the normal of $\mathbb{P}_1$. We first note that there are no $\mathbb{P}$-critical points in $\boldsymbol{x}_2$'s side of $\mathbb{P}_1$. Indeed, the state of the first layer is different than the state at $\boldsymbol{x}_1$, as the neuron corresponding to $\mathbb{P}_1$ changes its state. As it is assumed that each second layer critical hyperplane corresponds to a single neuron and single first layer state, it follows that there are no $\mathbb{P}$-critical points in $\boldsymbol{x}_2$'s side of $\mathbb{P}_1$. It is therefore enough to bound the probability that $\mathbf{x}$ is $\delta'$-close to $\boldsymbol{x}_1$'s side of $\mathbb{P}_1$, which is same as the probability that $\langle \mathbf{x} - \boldsymbol{x}_2, \boldsymbol{n}_1 \rangle \geq d(\boldsymbol{x}_2, \mathbb{P}_1) - \delta'$. Finally, $\langle \mathbf{x} - \boldsymbol{x}_2, \boldsymbol{n}_1 \rangle$ is a centered Gaussian with variance $\leq 1$. Hence, $p(\langle \mathbf{x} - \boldsymbol{x}_2, \boldsymbol{n}_1 \rangle \geq d(\boldsymbol{x}_2, \mathbb{P}_1) - \delta') \leq e^{-\frac{(d(\boldsymbol{x}_2, \mathbb{P}_1) - \delta')^2}{2}}$ $\square$

## D CORRECTNESS OF THE ALGORITHMS

**Lemma 12.** *Algorithm 2 reconstructs the correct affine transformation at an $\epsilon$-general point $\boldsymbol{x} \in \mathbb{R}^d$.*

*Proof.* Note that for any $\boldsymbol{y} \in \mathbb{R}^d$ with $\|\boldsymbol{y}\| \leq \epsilon$ we have,

$$
f(\boldsymbol{x} + \boldsymbol{y}) = f(\boldsymbol{x}) + \sum_{i=1}^{d} \frac{f(\boldsymbol{x} + \epsilon \boldsymbol{e}^{(i)}) - f(\boldsymbol{x})}{\epsilon} y_i.
$$

Hence, for every $z \in \mathbb{B}(x, \epsilon)$ we have

$$
\begin{aligned}
f(z) &= f(x + (z - x)) \\
&= f(x) + \sum_{i=1}^{d} \frac{f(x + \epsilon e^{(i)}) - f(x)}{\epsilon}(z_i - x_i) \\
&= \left( f(x) - \sum_{i=1}^{d} \frac{f(x + \epsilon e^{(i)}) - f(x)}{\epsilon} x_i \right) + \sum_{i=1}^{d} \frac{f(x + \epsilon e^{(i)}) - f(x)}{\epsilon} z_i.
\end{aligned}
$$

As $f$ is affine at $x$, we therefore get:

$$
w_i = \frac{f(x + \epsilon e^{(i)}) - f(x)}{\epsilon} \qquad and \qquad b = \left( f(x) - \sum_{i=1}^{d} \frac{f(x + \epsilon e^{(i)}) - f(x)}{\epsilon} x_i \right).
$$

$\square$

**Lemma 13.** *Algorithm 1 returns the left most critical point of a $\delta$-nice one-dimensional function $f$ in the range $(a, 1/\delta)$.*

*Proof.* Let $x^*$ is the left-most critical point in $(a, 1/\delta)$. Throughout the algorithm's execution, we have that $x_L < x^* < x_R$, as in each iteration, we choose the left half of the segment unless this half is affine (and therefore cannot have a critical point). As we start with a segment of size $2/\delta$ and split it two halves at each iteration, after $\lceil \log_2(2/\delta^2) \rceil$ iterations we left with $|x_L - x_R| < \delta$. Hence, in the final step, we have that $x_L$ is in the left-most piece, while $x_R$ is in the piece that is adjacent to the left-most piece. Therefore, $x^*$ is the point at the intersection of those two affine functions. If no critical point is in $(a, 1/\delta)$, then the segment is affine and we get that $\Lambda_L = \Lambda_R$.

Finally, note that all the points $x_L, x_R$ and $\frac{x_R + x_l}{2}$ during the execution of the algorithm are in the grid $\frac{2^{-\lceil \log_2(2/\delta^2) \rceil}}{\delta} \mathbb{Z}$ and therefore $\delta^2$-general. $\square$

**Lemma 14.** *Algorithm 3 returns critical hyperplane of $\delta$-non-degenerate critical point $x \in \mathbb{R}^d$, assuming the hyperplane is $\delta$ general.*

*Proof.* Let us assume that $x$ is a $\delta$-critical point of the $j$'th neuron. We will reconstruct the $j$'th neuron in two steps.

1. The first step is to find an affine function $\Lambda$ such that $\Lambda = \Lambda_{w_j, b_j}$ or $\Lambda = -\Lambda_{w_j, b_j}$. Let $\mathcal{M}'(x) := \mathcal{M}(x) - u_j \phi(\langle w_j, x \rangle + b_j)$. Note that $\mathcal{M}'$ is affine in $\mathbb{B}(x, \delta)$, as no neuron other than the $j$'th one changes its state in $\mathbb{B}(x, \delta)$. We have that in $\mathbb{B}(x, \delta)$ on one side of $x$'s critical hyperplane the network computes $\mathcal{M}'(x)$ and on the other hand it computes $\mathcal{M}'(x) + u_j(\langle w_j, x \rangle + b_j)$. Thus, to extract $\Lambda_{w_j, b_j}$ up to sign, we can simply compute the affine functions computed by the network on both sides of the $x$'s critical hyperplane, and subtract them.

2. The second step is to recover $u_j$. To this end, we note that if $u_j = 1$ then $\mathcal{M}(x)$ is strictly convex in $\mathbb{B}(x, \delta)$ as the sum of the affine function $\mathcal{M}'(x)$ and the convex function $u_j \phi(\langle w_j, x \rangle + b_j)$. Similarly, if $u_j = -1$ then $\mathcal{M}(x)$ is strictly concave in $\mathbb{B}(x, \delta)$. Thus, to recover $u_j$ we will simply check the convexity of $\mathcal{M}$ in $B(x, \delta)$ using algorithm 4.

Finally, note that $\phi(\Lambda(x))$ is either $\phi(\langle w_j, x \rangle + b_j)$ or $\phi(\langle w_j, x \rangle + b_j) - \langle w_j, x \rangle - b_j$. Hence, $u_j \phi(\Lambda(x))$ is either $u_j \phi(\langle w_j, x \rangle + b_j)$ or $u_j \phi(\langle w_j, x \rangle + b_j) - u_j \langle w_j, x \rangle - u_j b_j$. In particular, $u_j \phi(\Lambda(x))$ equals to $u_j \phi(\Lambda_{w_j, b_j}(x))$ up to an affine map. $\square$

# E   ON THE NON-ZERO PARTIAL DERIVATIVES ASSUMPTION

Consider a ReLU network

$$\mathcal{M}(\boldsymbol{x}) = \sum_{j=1}^{d_1} u_j \phi(\boldsymbol{w}_j \boldsymbol{x} + b_j) \tag{4}$$

and assume that for any $j$, $u_j \neq 0$ (otherwise the corresponding neuron can be dropped). We have that

$$\frac{\partial \mathcal{M}}{\partial x_i}(\boldsymbol{x}) = \sum_{j=1}^{d_1} u_j \phi'(\boldsymbol{w}_j \boldsymbol{x} + b_j) \boldsymbol{w}_j(i)$$

Now, if the weights are random, say that the $\boldsymbol{w}_j$'s are independent random variables such that $\boldsymbol{w}_j(i)$ has a continuous distribution, then w.p. 1, we have that for every non-zero vector $\boldsymbol{z} \in \{0, 1\}^{d_1}$, it holds that

$$\sum_{j=1}^{d_1} z_j \boldsymbol{w}_j(i) \neq 0$$

and hence $\frac{\partial \mathcal{M}}{\partial x_i}(\boldsymbol{x}) \neq 0$, unless the vector $\Lambda(\boldsymbol{x}) := (\boldsymbol{w}_1 \boldsymbol{x} + b_1, \ldots \boldsymbol{w}_{d_1} \boldsymbol{x} + b_{d_1})$ is in the negative orthant $\mathbb{R}^{d_1}_-$. It follows the non-zero partial derivatives assumption holds, provided if and only if the affine map $\Lambda$ maps the positive orthant $\mathbb{R}^d_+$ to the complement of the negative orthant $\mathbb{R}^{d_1}_-$. The following lemma shows that if $d_1 \gg d$, then this is often the case.

**Lemma 15.** *Assume that the pairs $(\boldsymbol{w}_j, b_j)$ are independent and symmetric[4], then*

$$p\left(\Lambda(\mathbb{R}^d) \cap \mathbb{R}^{d_1}_- \neq \emptyset\right) \leq \frac{\left(\frac{ed_1}{d}\right)^{d+1}}{2^{d_1}}$$

*Proof.* We first note that the number of orthants that has a non-negative intersection with $\Lambda(\mathbb{R}^d)$ is exactly the number of functions in the class

$$H = \left\{j \in [d_1] \mapsto \text{sign}(\boldsymbol{z}(j)) : \boldsymbol{z} \in \Lambda(\mathbb{R}^d)\right\}$$

Since $\Lambda(\mathbb{R}^d)$ is an affine space of dimension at most $d$, $H$ has VC dimension at most $d + 1$ (e.g. Anthony & Bartlet (1999)). Hence, by the Sauer-Shelah lemma (again, Anthony & Bartlet (1999))

$$|H| \leq \sum_{i=0}^{d+1} \binom{d_1}{i} \leq \left(\frac{ed_1}{d}\right)^{d+1}$$

Finally, since the $(\boldsymbol{w}_j, b_j)$'s and symmetric, the probability that $\Lambda(\mathbb{R}^d)$ intersects $\mathbb{R}^{d_1}_-$ is the same as the probability that it intersects any other orthant. Since there are $2^{d_1}$ orthants, and $\Lambda(\mathbb{R}^d)$ intersects at most $\left(\frac{ed_1}{d}\right)^{d+1}$ of them, it follows that $p\left(\Lambda(\mathbb{R}^d) \cap \mathbb{R}^{d_1}_- \neq \emptyset\right) \leq \frac{\left(\frac{ed_1}{d}\right)^{d+1}}{2^{d_1}}$.  $\square$

All in all we get the following corollary:

**Theorem 5.** *Assume that the pairs $(\boldsymbol{w}_j, b_j)$ are independent, symmetric, and has continuous marginals, w.p. $1 - \frac{\left(\frac{ed_1}{d}\right)^{d+1}}{2^{d_1}}$ we have that $\frac{\partial \mathcal{M}}{\partial x_i}(\boldsymbol{x}) \neq 0$ for all $\boldsymbol{x} \in \mathbb{R}^d$ and $i \in [d]$.*

---

[4]That is, for all $j \in [d_1]$, the distributions of $(\boldsymbol{w}_j, b_j)$ and $(-\boldsymbol{w}_j, -b_j)$ are the same.

