# OpenReview forum: "An Exact Poly-Time Membership-Queries Algorithm for Extracting a Three-Layer ReLU Network"
_ICLR.cc/2023/Conference — ICLR 2023 poster_

### Official Review · Reviewer_qJaS · 2022-10-23

**Confidence:** 3
**Correctness:** 3
**Technical Novelty And Significance:** 3
**Empirical Novelty And Significance:** Not applicable
**Recommendation:** 6

**Clarity, Quality, Novelty And Reproducibility:**

This paper is clearly written and well-organized. The assumptions and the idea of the algorithms/proofs are clearly addressed throughout the paper.


**Strength And Weaknesses:**

Strength

- The authors provide a polynomial-time query complexity algorithm for exact recovery of a two-layer neural networks with some general position assumptions.
- Additionally, an algorithm that can identify whether a neuron belongs to the first or the second layer is provided that is used in designing an algorithm for the recovery of a three-layer ReLU activated neural networks.
- The paper is well-written and organized in general and clearly conveys the main concepts and ideas for the recovery algorithms.

Weakness

- The proposed algorithms assume $\delta$-regular networks, satisfying 7 conditions from Definition 1, which are purely characteristics of the network itself, independent from either training dataset or learning algorithm. What is missing here is whether neural networks trained with gradient descent from random initialization can indeed be $\delta$-regular networks with high probability in practice. After training a simple 2 or 3 layer ReLU activated neural networks with some public datasets, can the authors indeed find that the conditions are well satisfied without the artificial weight perturbations?
- The authors provide a key concept in generalizing the recovery algorithm from 2-layer to 3-layer neural networks by identifying whether a neuron belongs to the first or the second layer. What is the main technical hurdle in extending the similar idea to networks with more depth? This discussion point is missing.
- Even though this work has value in theory as the authors argued, I think the algorithm should work in practice in recovering simple 2 or 3 layer neural networks. But, no experiment is provided by the author to judge the practical value of the proposed algorithms. Is there any reason one cannot actually use the proposed algorithm in real neural networks? Is it due to the strict assumptions imposed on the neural networks or the computational complexity?


**Summary Of The Paper:**

This paper considers the problem of learning a ReLU network from queries, which was recently remotivated by model extraction attacks. Assuming $\delta$-regular networks, the authors present a polynomial-time algorithm that can learn a depth-two ReLU network from queries and a polynomial-time algorithm that, with some additional assumptions, can learn a rich class of depth-three ReLU networks from queries.


**Summary Of The Review:**

This paper provides insightful results on designing algorithms for recovery of 2 or 3 layer neural networks, generalizing the previous results on this line of works. However, there are many assumptions, e.g. $\delta$-regular networks, imposed on the networks to guarantee the success of the proposed algorithms. Experiments or validation of the assumptions on practical neural networks are missing, which makes it hard to judge the practical value of the proposed algorithms.

---

### Official Review · Reviewer_vJNj · 2022-10-25

**Confidence:** 4
**Correctness:** 4
**Technical Novelty And Significance:** 3
**Empirical Novelty And Significance:** Not applicable
**Recommendation:** 6

**Clarity, Quality, Novelty And Reproducibility:**

The paper is overall rather well-written and contains novel ideas (especially
for reconstructing 3-layer networks.  The regularity assumption should perhaps
be incorporated in the main body as it is used in the algorithms presented
there.



Typos

Theorem 1 "There is an algorithm..., reconstruct" -> reconstructs

Section 2.3 "While approximated reconstructions"  -> approximate


**Strength And Weaknesses:**

Strengths

1. The paper gives the first polynomial-time algorithm for reconstructing
3-layer networks using membership queries.  The main challenge in
reconstructing 3-layer networks is identifying whether a critical hyperplane
comes from the first or second hidden layer (Algorithm 7).  The authors then
reduce this to reconstructing 2-layer networks.

Weaknesses

1. The reconstruction results require many assumptions (as the authors also
admit) (see Definition 1 of regular networks).  Fortunately, the authors show
that a (much easier to parse) sufficient condition is that the weights of the
network are perturbed by adding small uniform noise.  Still, it seems that at
least some of the assumptions are not necessary (even for exact recovery) and
are tailored to make the analysis easier.

2. I am not convinced that asking for exact network recovery is the ``right''
goal as it inherently leads to unnatural and perhaps impractical (especially
for deeper networks)  assumptions, e.g., having the exact output of the model
with zero error tolerance.



**Summary Of The Paper:**


This work considers the problem of exactly recovering ReLU networks using
membership queries, i.e., by having access to the exact output of the network
for any chosen input.  They give algorithms that can reconstruct 2 and 3 layer
ReLU networks in polynomial time under some assumptions on the weights of the
networks.

In particular, for 2-layer networks, this work shows that it is possible to
learn a network that is equivalent to the target network (in the sense that
the paper).  For 3-layer networks, they require additionally that the width of
the first layer is larger than the dimension of the input and that the top layer
has non-vanishing derivatives.  This work gives the first poly-time algorithm
the paper).



**Summary Of The Review:**


This work presents efficient algorithms for exact reconstruction of shallow
neural networks (up to 3-layers) using membership queries.  This work gives the
first polynomial time algorithm for reconstructing 3-layer networks and in this
sense, I believe it is a good first step toward deeper models.  Although I am not
convinced that exact reconstruction is the right direction (especially for
deeper models), I think the results of this work are above the threshold
for ICLR and I am leaning toward acceptance.

---

### Official Review · Reviewer_3xMF · 2022-10-27

**Confidence:** 2
**Clarity, Quality, Novelty And Reproducibility:** good
**Correctness:** 4
**Technical Novelty And Significance:** 3
**Empirical Novelty And Significance:** Not applicable
**Recommendation:** 5

**Strength And Weaknesses:**

The problem they solve is arguably rather far from the motivating context of reconstructing networks via attacks for at least a couple of reasons:
- The bound on the depth.
- The general position assumption.
It is for instance natural to have many nodes that apply some threshold to say $\sum x_i$. This would not be covered by their result.

Also it seems that the general position assumptions are necessitated by formulating the problem as an exact recovery problem as oppose dot functional recovery. The authors claim that exact recovery is essential for the motivating task of model extraction. I am not sure if I agree. If one can reconstruct a network that is equivalent to a large DNN by observing its behavior, why does it matter that the architecture is different?

It is a little unclear what the ultimate goal here would be. It is not clear if reconstruction algorithms for larger constant depths will shed light on the problem of how to reconstruct a real world neural net. Also, if the goal is to protect the IP of whoever designs the  neural net, perhaps negative results saying certain architectures are hard to reconstruct are more relevant?

On the other hand, their results are technically quite interesting and non-trivial, and are likely to generate followup work. I would not be opposed to accepting it for this reason, even if I am myself not very excited by the problem.

**Summary Of The Paper:**

This paper considers the problem of reconstructing a small-depth ReLU network with queries. They give results for depth 2 and depth 3 networks under some general position assumptions (and additional assumptions for depth 3). They solve the exact reconstruction problem where the goal is to reconstruct the parameters of the original network.



**Summary Of The Review:**

The paper is strong in terms of the technical results it proves. The motivation is a little weak. I lean towards reject, but would not object to accepting it if the paper has an enthusiastic backer.

---

### Decision · Program_Chairs · 2023-01-20

**Decision:**

Accept: poster

**Justification For Why Not Higher Score:**

The scores do not justify a higher score. While the contribution is solid, some of the assumptions imposed might not be necessary.

**Justification For Why Not Lower Score:**

Based on the reviews and my own reading, the proposed algorithm has nice ideas that may be useful in related context and in practice.

**Metareview: Summary, Strengths And Weaknesses:**

This paper gives the first polynomial-time algorithm that learns 3-layer neural networks with membership queries. Prior work could only handle depth-2 networks or imposed an extremely strong condition on the weights. While the current paper also makes some non-trivial assumptions (general position), these assumptions are significantly milder compared to prior work. Moreover, the algorithm has some nice ideas that may be useful in practice. Overall, it appears that the paper is somewhat above the acceptance threshold.

**Note From Pc:**

if the above contains the word "oral" or "spotlight" please see: "oral" presentation means -> notable-top-5% and "spotlight" means -> notable-top-25%. As stated in our emails, we are disassociating presentation type from AC recommendations